# IL-6 regulates autophagy and chemotherapy resistance by promoting BECN1 phosphorylation

Fuqing Hu[1,4], Da Song[1,4], Yumeng Yan[2], Changsheng Huang[1], Chentao Shen[1], Jingqin Lan[1], Yaqi Chen[1], Anyi Liu[1], Qi Wu[1], Li Sun[3], Feng Xu[1], Fayong Hu[1], Lisheng Chen[1], Xuelai Luo[1], Yongdong Feng[1], Shengyou Huang[2], Junbo Hu [1✉] & Guihua Wang [1✉]

Extracellular cytokines are enriched in the tumor microenvironment and regulate various important properties of cancers, including autophagy. However, the precise molecular mechanisms underlying the link between autophagy and extracellular cytokines remain to be elucidated. In the present study, we demonstrate that IL-6 activates autophagy through the IL-6/JAK2/BECN1 pathway and promotes chemotherapy resistance in colorectal cancer (CRC). Mechanistically, IL-6 triggers the interaction between JAK2 and BECN1, where JAK2 phosphorylates BECN1 at Y333. We demonstrate that BECN1 Y333 phosphorylation is crucial for BECN1 activation and IL-6-induced autophagy by regulating PI3KC3 complex formation. Furthermore, we investigate BECN1 Y333 phosphorylation as a predictive marker for poor CRC prognosis and chemotherapy resistance. Combination treatment with autophagy inhibitors or pharmacological agents targeting the IL-6/JAK2/BECN1 signaling pathway may represent a potential strategy for CRC cancer therapy.

[1] GI Cancer Research Institute, Tongji Hospital, Huazhong University of Science and Technology, Wuhan, P. R. China. [2] School of Physics, Huazhong University of Science and Technology, Wuhan, Hubei, P. R. China. [3] Department of Oncology, Tongji Hospital, Huazhong University of Science and Technology, Wuhan, P. R. China. [4] These authors contributed equally: Fuqing Hu, Da Song. ✉email: jbhu@tjh.tjmu.edu.cn; ghwang@tjh.tjmu.edu.cn

Cancer development is highly associated with the specific tumor microenvironment (TME), which includes stromal fibroblasts, infiltrating immune cells, the blood or lymphatic vascular networks, and the extracellular matrix[1–3]. Normal cells in the TME can be coopted or modified by cancer cells to produce a variety of growth factors, chemokines, and matrix-degrading enzymes that enhance the proliferation and invasion of cancer cells[4–6]. IL-6 is a common cytokine in the TME, and tumor-associated macrophages, granulocytes, fibroblasts, and cancer cells are all primary sources of IL-6[7]. IL-6 acts directly on cancer cells to trigger the expression of STAT3 target genes, the encoded proteins of which then drive cancer cell proliferation and survival, promoting angiogenesis, invasiveness, metastasis, and immunosuppression[8]. In addition, IL-6 stimulates the production of additional proinflammatory cytokines, as IL-6 recruits many types of immune cells into the TME[9,10]. Upregulated IL-6 levels are observed in patients with a variety of different cancers, such as breast[11], cervical[12], and colorectal cancer[13]. Remarkably, nuclear factor-κB (NF-κB) is a key transcription factor that drives the expression of IL-6[14], and hyperactivation of NF-κB is commonly observed in many human cancers[15]. Furthermore, circulating IL-6 levels are increased by surgery and chemoradiation[16,17] and are reported to be increased in patients with recurrent tumors[18]. Elevated serum IL-6 levels are also observed in patients with inflammatory bowel disease[19], and IL-6 levels generally correlate with tumor size, stage, and metastasis in colorectal cancer (CRC) patients[20]. Circulating IL-6 levels have been shown to be prognostic indicators of survival and predictors of a response to therapy in several types of cancer[21].

Autophagy is an intracellular lysosomal-dependent degradation system that typically engulfs and digests damaged organelles and long-lived proteins to provide nutrients or ATP for cell survival in response to extracellular and intracellular stress[22]. Autophagy plays a crucial role in several cellular functions, and its dysregulation is associated with tumorigenesis, tumor-stroma interactions, and resistance to cancer therapy. In cancer initiation and progression, substantial evidence has shown the dual role of autophagy as a tumor suppressor or a pro-oncogenic mechanism. In a normal cell, autophagy maintains normal cell homeostasis through the removal of oncogenic protein substrates, toxic unfolded proteins, and damaged organelles, which helps prevent chronic cellular damage and the transition into a cancer-initiating cell. However, once malignant cancers are established, increased autophagy enables tumor cell survival and chemotherapy resistance in many types of cancer tissues and in cancer recurrence, where autophagy is upregulated compared to normal adjacent tissues or primary cancer. Amino acid deprivation, hypoxia, growth factor deprivation and exposure to various chemicals, stress conditions, and some other types of stimuli are capable of activating autophagy[23]. In particular, the link between aberrant extracellular/intracellular signals and autophagy induction has been extensively studied to elucidate the precise mechanism underlying cancer-related autophagy[24]. As an important cytokine in the TME, autophagy enhances IL-6 release through NF-κB activation, and IL-6 promotes autophagy in many types of cancers[25]. However, the role of IL-6 in autophagy remains controversial, with conflicting reports regarding IL-6 function in the regulation of autophagy indicating that IL-6 regulation is dependent on the cell context[26].

In this study, we investigate how IL-6 regulates cancer chemotherapy resistance in an autophagy-dependent manner. We demonstrate that IL-6 induces the interaction between JAK2 and BECN1. Importantly, JAK2 directly phosphorylates BECN1 at Y333, leading to an enhanced BECN1-VPS34 interaction and autophagy, a mechanism by which IL-6 regulates cancer chemotherapy resistance. We further show that the level of BECN1 Y333 phosphorylation is a predictor of colorectal cancer patient outcome and that JAK2 inhibitor treatment combined with chemotherapy is more effective in inhibiting cancer cell growth. Thus, the results of the present study provide further insights into the molecular mechanism by which the TME functions in cancer therapy through IL-6/JAK2/BECN1-induced autophagy and show that BECN1 Y333 phosphorylation levels may serve as a predictor of chemotherapy resistance and a poor outcome in cancers, including CRC.

## Results

**IL-6 triggers cell autophagy via a STAT3-independent pathway.** To investigate the role of IL-6 in the regulation of autophagy, we treated cells with IL-6 and measured the levels of autophagy by examining the accumulation of LC3B-II. We showed that IL-6 induced the accumulation of LC3B-II in a dose- and time-dependent manner (Fig. 1a, b and Supplementary Fig. 1a, b, j). After treating cells stably expressing GFP-LC3B with IL-6, we observed that the number of green fluorescent puncta was strongly increased under IL-6 treatment (Fig. 1c and Supplementary Fig. 1c). Using electron microscopy, we observed more autophagosomes in IL-6-treated cells than in control cells (Fig. 1d, e). To determine the source of the increased number of autophagosomes mediated by IL-6 treatment, we examined the levels of free GFP (GFP protein freed from the fusion protein GFP-LC3B upon autophagic flux enhancement) and SQSTM1 (p62) expression in cancer cells transfected with a GFP-LC3B plasmid. Upon IL-6 treatment, the level of free GFP was upregulated, while the degradation of SQSTM1 was accelerated (Supplementary Fig. 1d), indicating that the IL-6 treatment-induced accumulation of autophagosomes resulted from the increased formation of autophagosomes rather than autophagic flux inhibition. To further assess the role of IL-6 in autophagy, we tested the effects of IL-6 on autophagic flux using the mCherry-GFP-LC3B plasmid with or without chloroquine (CQ, a lysosomal protease inhibitor used to inhibit autophagic flux). The data showed that IL-6 potently enhanced autophagic flux (the rate of red puncta formation in each cell), both in the absence and presence of CQ (Fig. 1f and Supplementary Fig. 1e). Taken together, these data indicate that IL-6 promotes the initiation of autophagosome formation and enhances autophagic flux in CRC.

STAT3 is the primary classic substrate signal protein regulated by IL-6, and previous studies have shown that STAT3 is a regulator of autophagy[27]. Thus, we assessed whether IL-6 regulates autophagy in a STAT3 signaling-dependent manner. We used a specific inhibitor of STAT3 to inhibit its activation and small interfering RNAs (siRNAs) to silence STAT3 expression in CRC cell lines; however, no change in the increased autophagy induced by IL-6 was observed (Fig. 1g, h and Supplementary Fig. 1f–i). Remarkably, we showed that IL-6 promotes autophagy in PC3 cell lines, in which endogenous STAT3 was deleted, indicating a role of IL-6 in autophagy that is independent of STAT3 that is consistent with the findings of a previous study using prostate cancer (PCa) cells[28] (Fig. 1i).

**IL-6 promotes cell autophagy in a JAK2 signaling-dependent manner.** Previous studies have shown the essential role of JAK2 in IL-6 downstream signaling. Given the important role of JAK2 in cancer development[29], we assessed whether JAK2 is involved in IL-6-induced autophagy. We employed a small molecular inhibitor of JAK2 kinase, CHZ868, and the data showed that CHZ868 decreased the accumulation of LC3B-II and inhibited the degradation of SQSTM1 in CRC cells following IL-6 treatment (Fig. 2a). Moreover, the increased number of green LC3B-II puncta induced by IL-6 were markedly decreased by

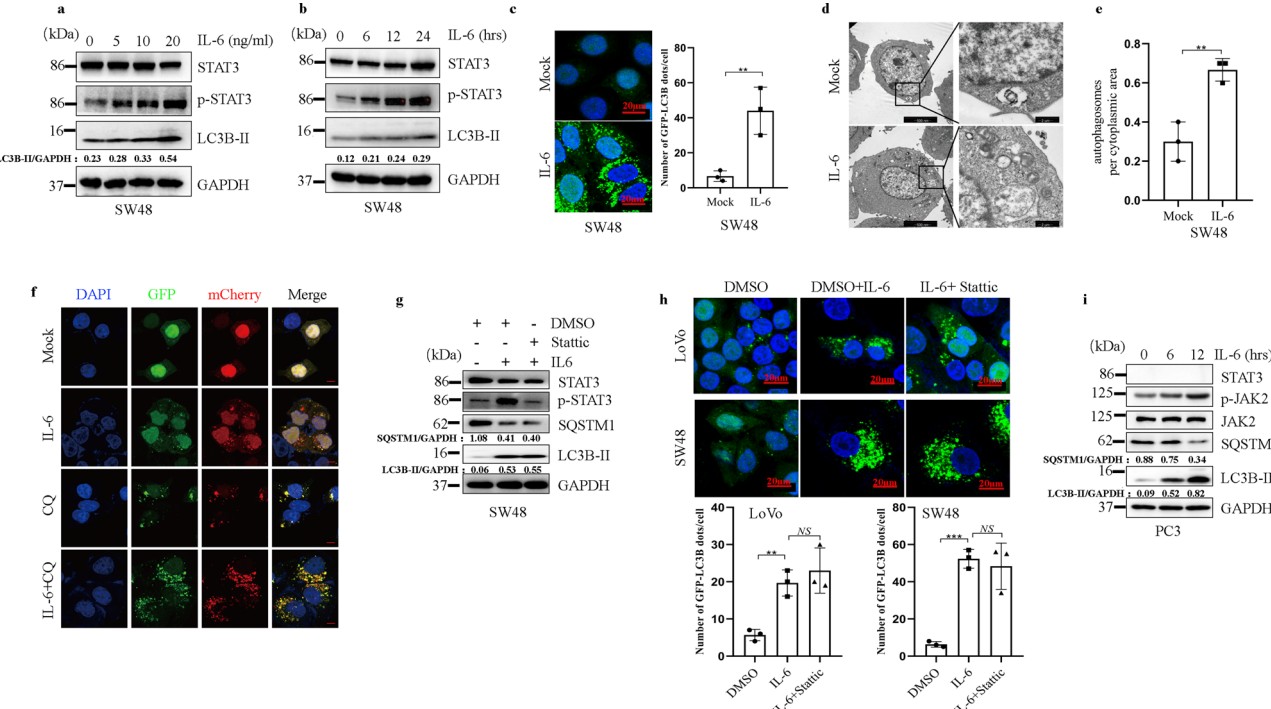

**Fig. 1 IL-6 triggers autophagy in cells via a STAT3-independent pathway. a, b** Exogenous IL-6 (20 ng/ml) promotes the accumulation of LC3B-II in SW48 cells treated with IL-6 in a dose- (**a**) and time-dependent (**b**) manner. Western blotting was performed to examine the expression of LC3B-II and SQSTM1. (*n* = 3 independent experiments). **c** Exogenous IL-6 (20 ng/ml) promotes the accumulation of GFP-LC3B puncta in SW48 cells. Immunofluorescence (IF) staining analyses of GFP-LC3B3 puncta in SW48 cells treated with IL-6 (20 ng/ml) for 12 h. Quantitative analysis of GFP-LC3B puncta is shown in the right panel. **P < 0.01. Scale bars, 20 μm (*n* = 3 independent experiments). **d, e** Representative images from transmission electron microscopy showing autophagosomes (arrows) in SW48 cells after treatment with IL-6 (20 ng/ml) (**d**). Quantitative analysis of autophagosomes is shown in the right panel (**e**). **P < 0.01. Scale bars, 5 μm (*n* = 3 independent experiments). **f** Examination of autophagic flux with the mCherry-GFP-LC3 reporter in SW48 cells. SW48 cells stably expressing the mCherry-GFP-LC3B fusion protein were separately treated with IL-6 (20 ng/ml) in the absence or presence of CQ (25 μM). Confocal microscopy images are shown. Scale bars, 20 μm (*n* = 3 independent experiments). **g** Western blotting was performed for SW48 cells separately treated with IL-6 (20 ng/ml) in the absence or presence of Stattic (20 μM) for 12 h (*n* = 2 independent experiments). **h** SW48 and LoVo cells stably expressing GFP-LC3B fusion protein were separately treated with IL-6 (20 ng/ml) in the absence or presence of Stattic (20 μM) for 12 h. Confocal microscopy images are shown. Quantitative analysis of GFP-LC3B puncta is shown in the lower panels. **P < 0.01, ***P < 0.001. Scale bars, 20 μm (*n* = 3 independent experiments). **i** Exogenous IL-6 promotes the accumulation of LC3B-II and the degradation of SQSTM1 in PC3 cells treated with IL-6 in a dose-dependent manner for 12 h (*n* = 2 independent experiments). In **c**, **e**, and **h**, the values are presented as the means ± s.d.; *p* values (Student's *t* test, two-sided) with comparisons made to the control or different indicated groups are shown. Source data are provided in the Source Data file.

pharmacological inhibition of JAK2 kinase activity (Fig. 2b and Supplementary Fig. 2a, b). To further investigate the role of JAK2 in autophagy induced by IL-6, we evaluated the effects of JAK2 overexpression and silencing of JAK2 on autophagy induced by IL-6. We observed that JAK2 overexpression enhanced the formation of autophagosomes and autophagic flux in CRC cells after IL-6 treatment (Fig. 2c, d and Supplementary Fig. 2c, d). In addition, both Western blot and immunofluorescence results showed that JAK2 knockdown significantly blocked IL-6-induced autophagy in CRC cells (Fig. 2e, f and Supplementary Fig. 2e, f). Taken together, these data indicate that IL-6 induces autophagy in a JAK2 signaling-dependent manner.

**JAK2 interacts with and phosphorylates BECN1 at Y333.** As an important nonreceptor tyrosine kinase protein, JAK2 phosphorylates multiple types of substrate proteins that play crucial roles in regulating biological behavior[30]. Our results indicated that IL-6 induces autophagy in a JAK2- not STAT3 signaling-dependent manner suggesting that JAK2 may bind to a protein that can facilitate IL-6-induced autophagy. To test our hypothesis, we performed online data analysis (https://string-db.org/) and observed that BECN1, a crucial regulator of autophagy formation,

may function as a JAK2 substrate protein (Supplementary Fig. 3a). We confirmed the interaction between endogenous JAK2 and BECN1 by using a coimmunoprecipitation assay (Fig. 3a, b and Supplementary Fig. 3b–d), which was notably promoted by IL-6 stimulation (Fig. 3c and Supplementary Fig. 3e). Similar results were also observed in the immunofluorescence assay, where more endogenous BECN1 colocalized with JAK2 in the cytoplasm upon IL-6 stimulation (Fig. 3d). Subsequently, we attempted to precisely determine the molecular interaction between JAK2 and BECN1. To this end, a series of JAK2 and BECN1 mutants were constructed and transfected into 293T cells (Supplementary Fig. 3f), the subsequent analysis of which demonstrated that the JH-1 domain of JAK2 is responsible for the interaction between JAK2 and BECN1 (Supplementary Fig. 3g). In addition, we observed that the evolutionarily conserved domain (ECD) of the BECN1 protein was sufficient to interact with JAK2 (Supplementary Fig. 3h). These data show that the JH-1 domain of JAK2 and the ECD domain of BECN1 are indispensable for JAK2/BECN1 complex formation.

Since JAK2 is a known tyrosine kinase that phosphorylates substrates and interacts with BECN1, we examined whether BECN1 is a substrate of JAK2. Notably, we observed that IL-6 treatment-induced BECN1 phosphorylation (Supplementary Fig. 4a), and

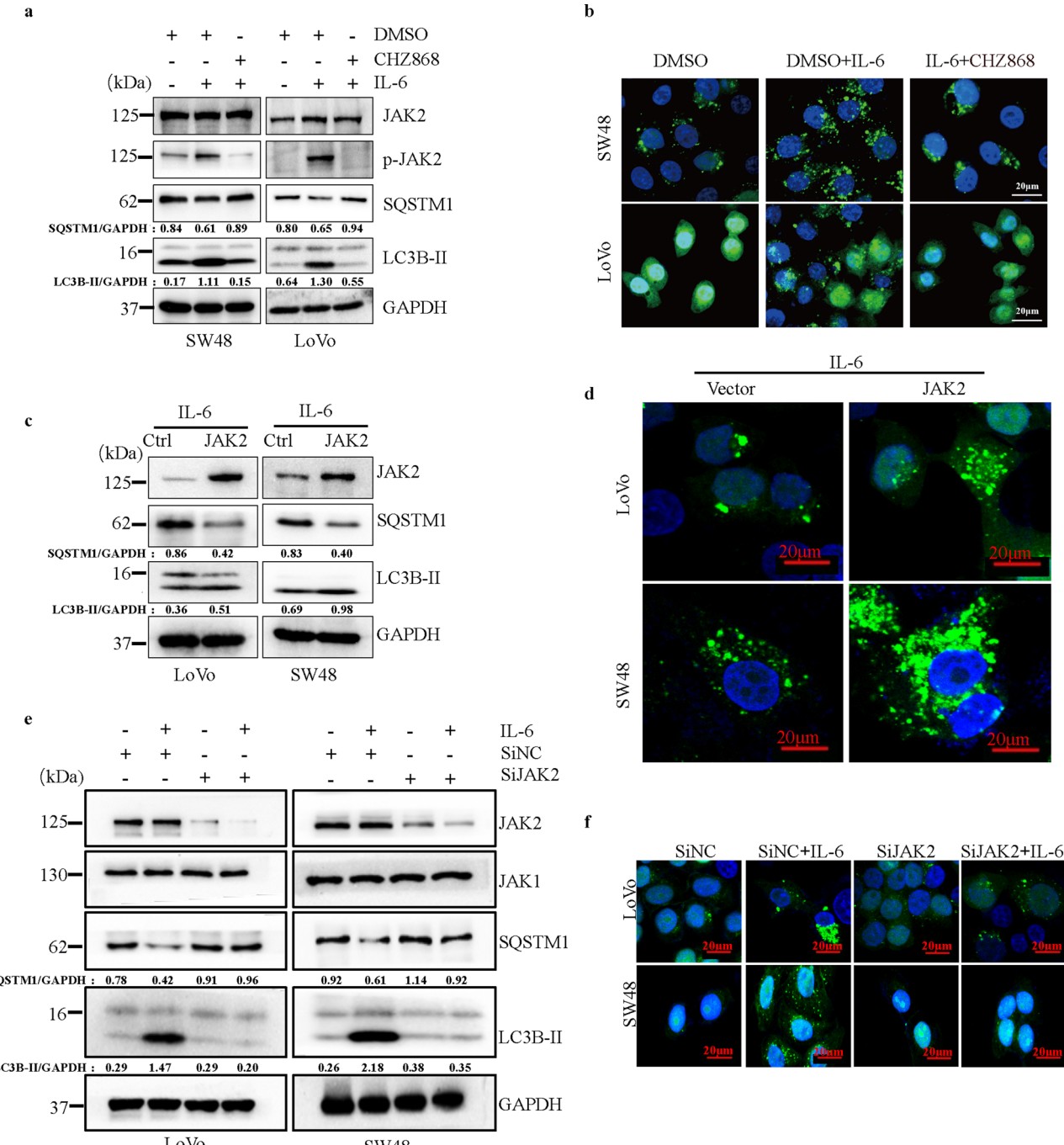

**Fig. 2 IL-6 promotes autophagy in a JAK2 signaling-dependent manner. a** Western blotting was performed for SW48 and LoVo cells separately treated with IL-6 (20 ng/ml) in the absence or presence of CHZ868 (0.2 μM) for 12 h (n = 2 independent experiments). **b** SW48 and LoVo cells stably expressing the GFP-LC3B fusion protein were separately treated with IL-6 (20 ng/ml) in the absence or presence of CHZ868 (0.2 μM) for 12 h. Confocal microscopy images are shown. Scale bars, 20 μm (n = 3 independent experiments). **c** SW48 and LoVo cells were separately transfected with the JAK2 or control plasmid. After transfection for 24 h, cells were treated with IL-6 (20 ng/ml) for 12 h. Western blotting was performed to examine the expression of LC3B-II and SQSTM1 (n = 2 independent experiments). **d** SW48 and LoVo cells stably expressing the GFP-LC3B fusion protein were separately transfected the JAK2 or control plasmid, and after stimulation with IL-6 (20 ng/ml) for 12 h, confocal microscopy images were obtained to measure the number of GFP puncta. Scale bars, 20 μm (n = 3 independent experiments). **e** SW48 and LoVo cells were separately transfected with small interfering RNA targeting JAK2 (SiJAK2) or small interfering RNA targeting a negative control gene (SiNC). After transfection for 24 h and following stimulation with IL-6 (20 ng/ml) for 24 h, western blotting was performed to examine the expression of LC3B-II and SQSTM1 (n = 2 independent experiments). **f** SW48 and LoVo cells stably expressing GFP-LC3B fusion protein were separately transfected with SiJAK2 and SiNC. After stimulation with IL-6 (20 ng/ml) for 24 h, confocal microscopy images were obtained to measure the number of GFP puncta. Scale bars, 20 μm (n = 3 independent experiments). Source data are provided in the Source Data file.

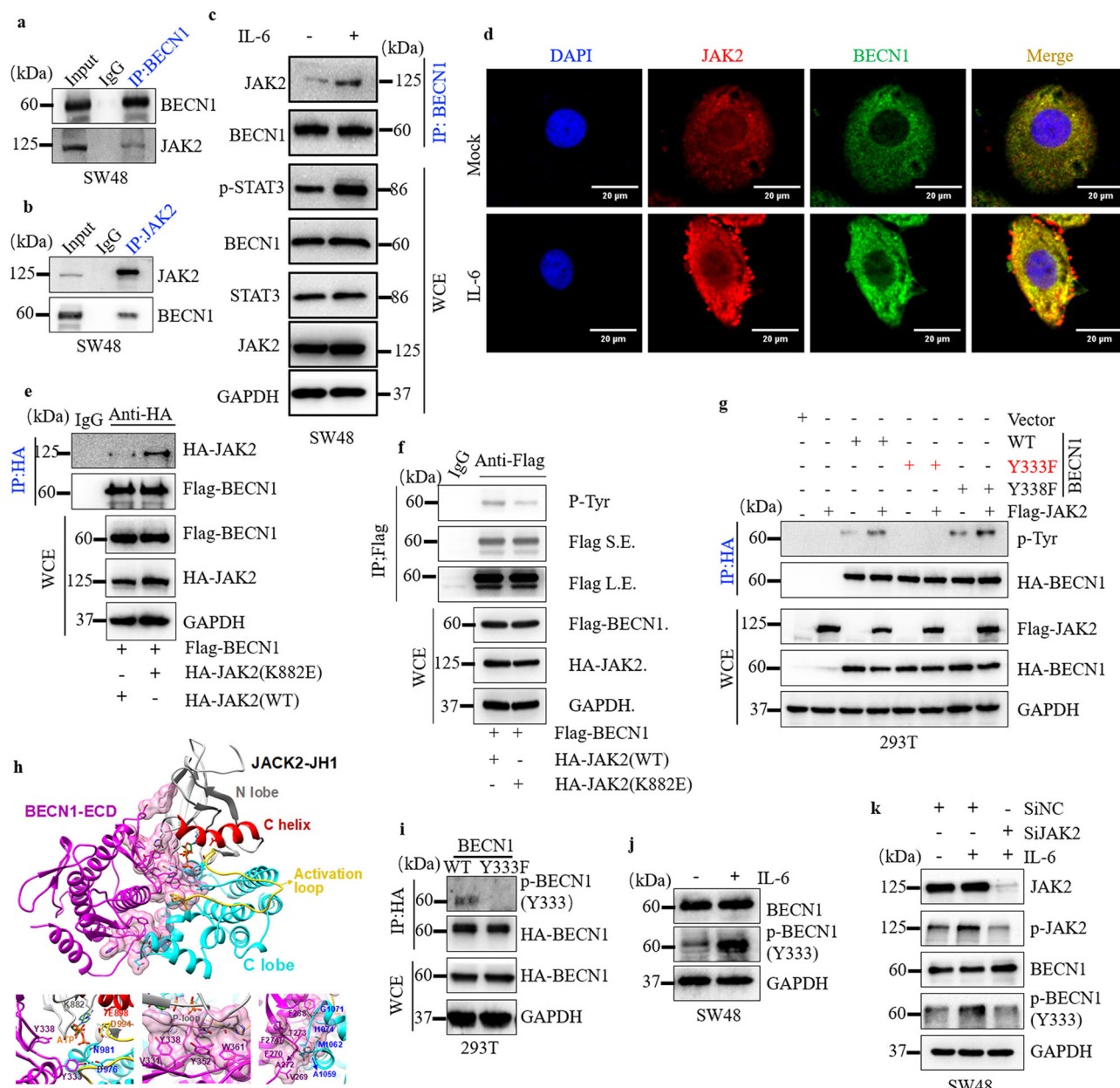

**Fig. 3 JAK2 interacts with BECN1 and phosphorylates BECN1 at Y333. a, b** Immunoprecipitation (IP) analyses were performed to examine the interaction between BECN1 and JAK2 (**a**) or JAK2 and BECN1 (**b**) in SW48 cells. **c** Co-IP was performed to examine the relationship between JAK2 and BECN1 in SW48 cells in the absence or presence of IL-6 (20 ng/ml) for 12 h (upper panels). Western blotting was performed on whole-cell extracts (WCEs) (lower panels). **d** Immunofluorescent staining (IF) analyses of LoVo cells using anti-BECN1 and anti-JAK2 antibodies. Yellow puncta, double-stained BECN1 and JAK2. Scale bars, 20 μm. **e** Co-IP was performed to examine the interaction of JAK2 and BECN1 in HEK293T cells cotransfected with Flag-BECN1 and HA-JAK2 (WT)/HA-JAK2 (K882E) (upper panels) for 24 h. Western blotting was performed on WCEs (lower panels). **f** Co-IP was performed to examine the tyrosine phosphorylation of Flag-BECN1 in HEK293T cells cotransfected with Flag-BECN1 and HA-JAK2 (WT)/HA-JAK2 (K882E) (upper panels) for 24 h. Western blotting was performed on WCEs (lower panels). **g** Co-IP analyses for HA-BECN1 and p-Tyr in HEK293T cells expressing Flag-JAK2, a vector control (Vector), HA-BECN1 WT, HA-BECN1 Y333F, or HA-BECN1 Y338F. Western blotting was performed on WCEs (lower panels). **h** Modeled complex structure between the ECD domain (magenta, left) and JH1 domain (right), where the JH1 domain consists of four regions: an N-terminal lobe (gray), a C-terminal lobe (cyan), a C helix (red), and an activation loop (yellow). ATP is colored by element, with carbon atoms in green, oxygen atoms in red, nitrogen atoms in blue, and phosphorus atoms in orange. The residues involved in the hydrophobic interaction are shown in pink sphere representation, with atoms shown as sticks. The hydrogen bond is shown by black dashed lines. On the bottom, from left to right, are the enlarged images for the interface in the catalytic site, the interface between the β sheet with the phosphorylation site of the ECD domain and the N-terminal lobe of the JH1 domain, and the interface between the hydrophobic loop of the ECD domain and the C-terminal lobe of the JH1 domain, respectively. **i** IP analyses for HA-BECN1 and p-BECN1 (Y333) in HEK293T cells expressing HA-BECN1 WT or HA-BECN1 Y333F. Western blotting was performed on WCEs (lower panels). **j** Western blotting was performed to examine the expression of BECN1 and p-BECN1 (Y333) in LoVo cells in the absence or presence of IL-6 (20 ng/ml) for 12 h. **k** SW48 cells were separately transfected with SiJAK2 and SiNC. After transfection for 48 h and following stimulation with IL-6 (20 ng/ml) for 12 h, western blotting was performed. Western blots are representative of two independent experiments. WCE whole-cell extract. Source data are provided in the Source Data file.

JAK2 knockdown impaired the increased tyrosine phosphorylation levels of BECN1 upon stimulation with IL-6 (Supplementary Fig. 4b). In addition, kinase-inactive JAK2 mutant (K882E) and wild-type JAK2 (WT) constructs were generated and transfected into 293 T cells. Co-IP results showed a increased interaction between the kinase-inactive JAK2 mutant and BECN1 compared to wild-type JAK2 (Fig. 3e). Moreover, phosphorylation assessments showed that the kinase-inactive JAK2 mutant had only a slight effect on the level of BECN1 phosphorylation compared to wild-type JAK2 (Fig. 3f). To identify the site in BECN1 phosphorylated by JAK2, we used a public database to predict the potential phosphorylation sites via a multidatabase analysis. Two candidate tyrosines (Y333 and Y338) in BECN1 potentially phosphorylated by JAK2 were selected for analysis (Supplementary Fig. 4c). To further confirm that BECN1 is phosphorylated by JAK2, site-directed mutant constructs of BECN1 (Y333F and Y338F, which are inactive nonphosphorylatable mutants) and Flag-JAK2 were transfected into 293T cells. Co-IP assay results showed that JAK2 markedly augmented the tyrosine phosphorylation levels of wild-type BECN1 and the phosphorylation-defective BECN1 Y338F mutant. However, the BECN1 Y333F mutant did not show altered phosphorylation levels and was barely affected by JAK2 (Fig. 3g). We also did not observe differences in the interactions between JAK2 and the wild-type BECN1 or with the phosphorylation-defective mutant BECN1 Y333F (Supplementary Fig. 4d). In addition, we used computational modeling of the interaction between BECN1 and JAK2 to determine why Y333 of BECN1 is phosphorylated by JAK2. The results presented in Fig. 3h show the modeled structure of the ATP-bound JH1 domain of JAK2 in complex with the ECD domain of BECN1, where the left part of the top Fig. is the ECD domain, and the right part is the JH1 domain. We also observed that the interaction between the two domains results from three regions in the complex structure: the interface around the catalytic site, the β-sheet with the Y333 phosphorylation site in the ECD domain interacting with the N-terminal lobe of the JH1 domain (top interface), and a hydrophobic loop of the ECD domain interacting with the C-terminal lobe of the JH1 domain (bottom interface). Enlarged images of the three regions involved in the JAK2 and BECN1 interaction are presented in Fig. 3h. For the catalytic site, similar to other tyrosine kinases[31,32], residue Y333 of the ECD domain is close to the conserved residue D976 and forms a hydrogen bond to help stabilize and locate the tyrosine to be phosphorylated. The conserved residues E898, K882, D994, and N981 form the binding pocket to bind the substrate ATP molecule (lower left panel of Fig. 3h). For the top interface, four hydrophobic residues in the ECD domain (V331, Y338, Y352, and W361) form a hydrophobic surface interaction with the glycine-rich P-loop of the N-terminal lobe of the JH1 domain (lower middle panel of Fig. 3h). For the bottom interface, four hydrophobic residues in two helixes of the JH1 domain (G1071, I1074, M1062, and A1059) form a hydrophobic pocket, and the hydrophobic loop, containing V269, F270, T273, and F274, of the ECD domain interacts with the pocket to form a hydrophobic interface (lower right panel of Fig. 3h). The model also shows that Y333 of the ECD is at the terminal region of the β-sheet, while Y338 is located in the middle. Therefore, Y333 can be inserted into the catalytic site close to the key catalytic residues, whereas it is difficult for Y338 to enter the binding site because of steric clashes. These results suggest that phosphorylation occurs at Y333 rather than Y338 of BECN1. Taken together, these data indicate that JAK2 specifically phosphorylates BECN1 at Y333. To further assess whether BECN1 Y333 is a JAK2-specific phosphorylation site, we generated an antibody that specifically detects p-Y333 in BECN1. Dot blots and endogenous detection showed that the antibody against p-BECN1 (Y333) was a specific antibody (Fig. 3i and Supplementary Fig. 4e). Consistent with our findings, IL-6 stimulation increased p-BECN1 (Y333) levels (Fig. 3j).

Importantly, JAK2 knockdown significantly abolished the increased levels of p-BECN1 (Y333) induced by IL-6 stimulation in both SW48 and LoVo cells (Fig. 3k and Supplementary Fig. 4f). Taken together, our data strongly suggest that IL-6 exerts its proautophagic function in CRC by regulating the JAK2-BECN1 interaction and BECN1 tyrosine phosphorylation at Y333.

**BECN1 Y333 phosphorylation is required for IL-6-induced autophagy.** BECN1 is an allosteric modulator of class III phosphatidylinositol 3-kinase (PI3KC3) complexes (PI3KC3-C1 and PI3KC3-C2) and is crucial for autophagy. PI3KC3-C1 functions in autophagic vesicle enucleation at the initial stages of autophagosome formation and comprises BECN1, VPS34, VPS15, ATG14, and Ambra, while PI3KC3-C2 is involved in autophagolysosomal maturation and comprises BECN1, VPS34, VPS15, and UVRAG[33]. To evaluate the physiological role of the IL-6-induced phosphorylation of BECN1 Y333, we first assessed the interaction of BECN1 with its binding partners, including its negative regulators, such as BCL-2 and Rubicon, and its positive regulators, such as VPS34 and VPS15 (two PI3KC3 components that bind to the ECD domain of BECN), under IL-6 treatment. We showed that the interaction between endogenous BECN1 and VPS34 or VPS15 was enhanced upon IL-6 stimulation, while the interaction between BECN1 and BCL-2 or Rubicon were abolished, whereas IL-6 had no effect on the interaction between BECN1 and UVRAG or ATG14 (Fig. 4a). Pharmacological inhibition of JAK2 remarkably abolished the interaction between endogenous BECN1 and VPS34 or VPS15 and promoted the interaction between BECN1 and BCL-2 or Rubicon (Fig. 4a). To further assess whether IL-6-induced p-BECN1 (Y333) affects the formation of the BECN1 interactome, we transfected site-directed mutants of BECN1 (Y333F and Y333E, phosphorylation-defective and phospho-mimetic mutants of BECN1, respectively) or wild-type BECN1 (WT). Co-IP assay results demonstrated that the phospho-mimetic BECN1 mutant increased the interaction between VPS15 and VPS34 and reduced the interaction between BCL-2 and Rubicon. Conversely, the phosphorylation-defective BECN1 mutant had no effect on the interaction between VPS15 and VPS34 compared to WT BECN1 (Fig. 4b), suggesting that IL-6-induced p-BECN1 (Y333) plays a crucial role in BECN1 interactome formation.

Given the important role of p-BECN1 (Y333) in the formation of the BECN1 interactome, we further investigated the role of p-BECN1 (Y333) in IL-6-induced autophagy. We used MCF-7 human breast carcinoma cells as a research model because of the low amounts of endogenous BECN1 observed in these cells compared to other cell lines (Supplementary Fig. 5a). Upon IL-6 stimulation, the accumulation of LC3B-II and the rate of SQSTM1 degradation in cells transfected with the BECN1 Y333F mutant construct was impaired compared to cells transfected with the WT BECN1 construct (Fig. 4c). Furthermore, cells transfected with the BECN1 Y333E mutant construct had higher basal autophagy than those transfected with the WT BECN1 construct (Fig. 4c). Meanwhile, IL-6 significantly induced autophagy in cells transfected with the WT BECN1 construct or in untransfected cells (Fig. 4c), strongly suggesting that IL-6-induced autophagy is dependent on p-BECN1 (Y333). To further investigate the role of IL-6-induced BECN1 phosphorylation in autophagy, BECN1 knockout LoVo cell lines (KO-BECN1-LoVo) were constructed using CRISPR/Cas9 technology (Supplementary Fig. 5b). Western blot results and Sanger sequencing analysis of genomic DNA confirmed the generation of BECN1 knockout LoVo cell lines (Supplementary Fig. 5b), after which KO-BECN1-LoVo cells were stably transfected with BECN1 mutant constructs. The accumulation of LC3B-II and the rate of SQSTM1

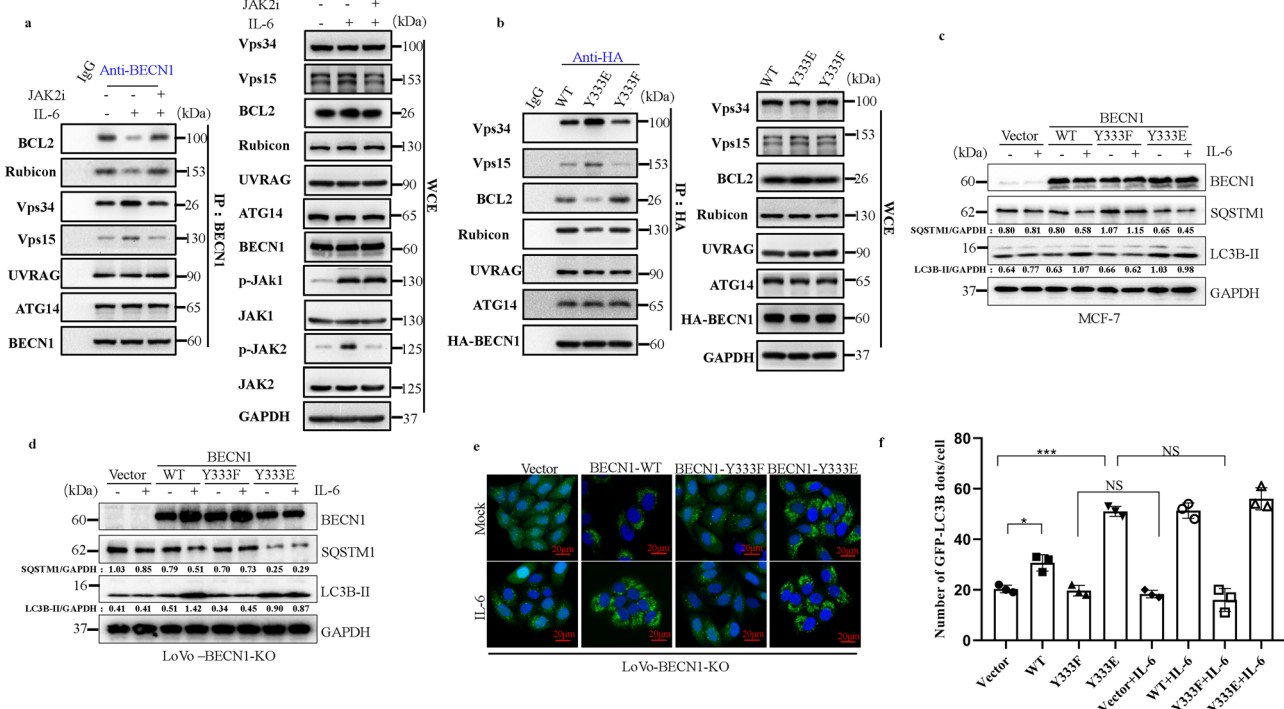

**Fig. 4 BECN1 Y333 phosphorylation is required for IL-6-induced autophagy. a** IP analyses for VPS34, VPS15, BCL2, Rubicon, UVRAG, ATG14, and BECN1 in HEK293T cells in the absence or presence of IL-6 (20 ng/ml) for 12 h or in the absence or presence of CHZ868 (0.2 μM) treatment for 12 h. Western blotting was performed on WCEs (lower panels). **b** Co-IP analyses for HA-BECN1, VPS15, BCL2, Rubicon, UVRAG, ATG14, and VPS34 in HEK293T cells expressing HA-BECN1 WT, HA-BECN1 Y333F, or HA-BECN1 Y333E. Western blotting was performed on whole-cell lysates (lower panels). **c** Western blot analyses for BECN1, SQSTM1, and LC3B-II in MCF-7 cells expressing a vector control (Vector), HA-BECN1 WT, HA-BECN1 Y333F, or HA-BECN1 Y333E in the absence or presence of IL-6 (20 ng/ml) for 12 h. **d** Western blot analyses for BECN1, SQSTM1, LC3B-II in LoVo-BECN1-KO cells expressing a vector control (Vector), HA-BECN1 WT, HA-BECN1 Y333F, or HA-BECN1 Y333E in the absence or presence of IL-6 (20 ng/ml) for 12 h. **e, f** IF analyses for GFP-LC3B3 puncta in LoVo-BECN1-KO cells expressing a vector control (Vector), HA-BECN1 WT, HA-BECN1 Y333F or HA-BECN1 Y333E in the absence or presence of IL-6 (20 ng/ml) for 12 h. Confocal microscopy images are shown ($n = 3$ independent experiments) (**f**). Quantification of the number of GFP-LC3B puncta is shown in the right panel (**g**). *$P < 0.05$, ***$P < 0.001$. NS not significant. Scale bars, 20 μm. The values are presented as the mean ± s.d.; $p$ values (Student's $t$ test, two-sided) with comparisons made to the control or different indicated groups are shown. Western blots are representative of two independent experiments. WCE whole-cell extract. Source data are provided in the Source Data file.

degradation in KO-BECN1-LoVo cells transfected with the BECN1 Y333F mutant construct was inhibited compared to cells transfected with the WT BECN1 construct (Fig. 4d). However, KO-BECN1-LoVo cells transfected with the BECN1 Y333E mutant showed high levels of autophagy (Fig. 4d), and IL-6 treatment had only a slight effect on these cells (Fig. 4d). Immunofluorescence (IF) analysis also yielded similar results (Fig. 4e, f). Taken together, our data indicate that IL-6 induces the JAK2-mediated phosphorylation of BECN1 and promotes autophagy by regulating the BECN1 interactome.

**IL-6-induced BECN1 Y333 phosphorylation promotes cancer chemotherapy resistance.** Elevated IL-6 levels are observed in patients with different types of cancer, such as breast, cervical, colorectal, and nonsmall-cell lung cancer (NSCLC). Furthermore, IL-6 levels are correlated with tumor size, stage, and metastasis in patients with CRC. In addition, IL-6 levels have been shown to be prognostic indicators of survival and predictors of a response to therapy in several different types of cancer. To evaluate the role of IL-6-induced BECN1 Y333 phosphorylation in cancer therapy, we first used a CRC model and assessed the potential role of IL-6 in CRC chemotherapy resistance. We showed that IL-6 stimulation significantly abolished the decreased cell viability induced by oxaliplatin (OXA) and 5-fluorouracil (5-Fu) (Fig. 5a and Supplementary Fig. 6a). OXA and 5-Fu markedly activated caspase-3/PARP apoptotic signaling, which could be inhibited by

simultaneous treatment with IL-6 (Fig. 5b and Supplementary Fig. 6b), indicating that IL-6 promotes chemotherapy resistance in CRC cells. To further evaluate whether IL-6 regulates chemotherapy sensitivity in CRC through the modulation of autophagy, we treated cells with IL-6 and chemotherapy drugs in the presence of CQ, a classic inhibitor of autophagy that blocks autophagosome binding to lysosomes by altering the acidic environment of lysosomes. Interestingly, we observed that the IL-6-induced increase in cell viability was impaired by CQ treatment (Fig. 5c and Supplementary Fig. 6c). Furthermore, the inhibition of caspase-3/PARP apoptotic signaling induced by IL-6 was re-enhanced by CQ treatment in CRC cell lines (Fig. 5d and Supplementary Fig. 6d). Furthermore, we observed that IL-6 decreased cell apoptosis induced by chemotherapy drug treatment, and the effects were impaired by the inhibition of autophagy (Supplementary Fig. 6e). To further assess the autophagy-mediated role of IL-6 in chemotherapy resistance, we performed an in vivo experiment in which CT26 cells were subcutaneously injected into BALB/c mice to generate CRC xenograft models. Following treatment with IL-6, CQ, and chemotherapy drugs, the tumor growth rate and tumor weight were measured to assess the role of IL-6-mediated autophagy in chemotherapy resistance. We showed that chemotherapy drugs significantly inhibited growth rate and weight of tumors, while IL-6 treatment abolished the inhibitory effects of chemotherapy drugs (Fig. 5e, f and Supplementary Fig. 6f). Moreover, CQ treatment blocked IL-6-induced

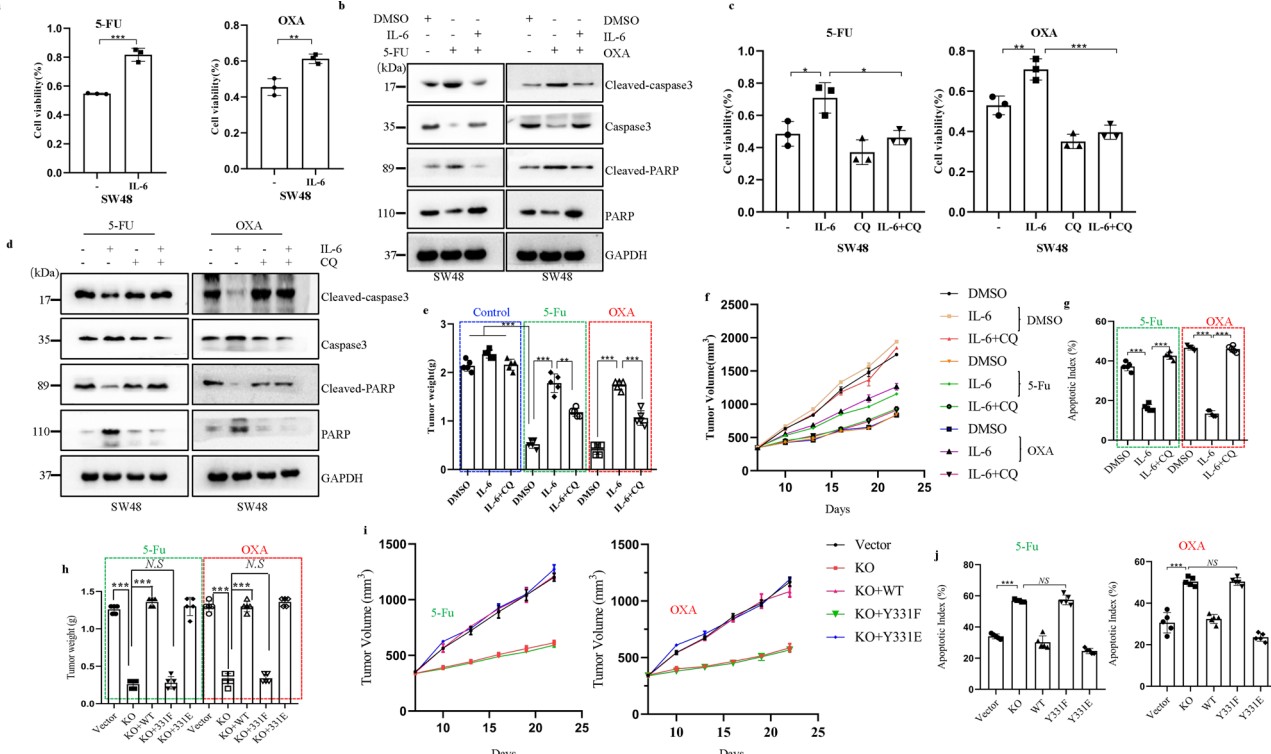

**Fig. 5 IL-6-induced BECN1 Y333 phosphorylation promotes cancer chemotherapy resistance. a** CCK-8 assays for cell viability of SW48 cells separately pretreated with DMEM or IL-6 (20 ng/ml) for 8 h in the presence of DMSO, 5-Fu (800 μM), or OXA (100 μM) for 36 h. **P < 0.01, ***P < 0.001 (n = 3 independent experiments). **b** Western blot analyses for cleaved caspase-3 and cleaved PARP1 in SW48 cells separately pretreated with DMEM or IL-6 (20 ng/ml) for 8 h in the presence of DMSO, 5-Fu (800 μM) or OXA (100 μM) for 36 h (n = 3 independent experiments). **c** CCK-8 assays for cell viability of SW48 cells separately pretreated with DMEM or IL-6 (20 ng/ml) for 8 h in the presence of CQ (25 μM), 5-Fu (800 μM) or OXA (100 μM) for 36 h. *P < 0.05, **P < 0.01, ***P < 0.001 (n = 3 independent experiments) **d** Western blot analyses for cleaved caspase-3 and cleaved PARP1 in SW48 cells separately pretreated with DMEM or IL-6 (20 ng/ml) for 8 h in the presence of CQ (25 μM), 5-Fu (800 μM), or OXA (100 μM) for 36 h. (n = 3 independent experiments). **e** In vivo CRC xenografts derived from CT26 cells were treated with IL-6, 5-Fu, or OXA. Quantification of tumor weights from the different CRC xenografts is shown. **P < 0.01, ***P < 0.001. **f** Tumor volumes at the experimental endpoint were quantified in CT26 cell xenografts. **g** The apoptotic index was quantified for the different CRC xenografts. **P < 0.01, ***P < 0.001. **h–j** In vivo CRC xenografts derived from CT26 cells and CT26-Becn1-KO cells separately expressing a vector control (Vector), Becn1 WT, Becn1 Y3331F, or Becn1 Y3331 were treated with DMSO, 5-Fu or OXA. Tumor weights (**h**), tumor volumes (**i**), and apoptotic index (**j**) quantified at the experimental endpoint are shown (n = 5 mice per genotype). **P < 0.01, ***P < 0.001. NS not significant. In (**a**, **c**, **e**, **g**, **h** and **j**), the values are presented as the means ± SEM; p values (Student's t test, two-sided) with control or different indicated groups are shown. Source data are provided in the Source Data file.

chemotherapy resistance with respect to the growth rate and weight of tumors (Fig. 5e, f and Supplementary Fig. 6f). Immunohistochemistry (IHC) results also showed that IL-6 inhibited activation of caspase-3/PARP apoptotic signaling and that CQ treatment potentially reversed the protective role of IL-6 in the apoptotic signaling pathway (Fig. 5g and Supplementary Fig. 6g). Taken together, these results demonstrate that IL-6 promotes chemotherapy resistance via activation of the autophagy pathway.

As we showed that IL-6 contributes to CRC autophagy through the regulation of p-BECN1 (Y333), we further evaluated the role of p-BECN1 (Y333) in CRC chemotherapy resistance. To this end, we generated LoVo cell lines expressing WT, Y333F, or Y333E BECN1. Cell colony formation assays showed that cell growth was strongly inhibited in LoVo-BECN1-KO cells following treatment with chemotherapy drugs (Supplementary Fig. 7a, b). Interestingly, complementation with WT or Y333E BECN1 in LoVo-BECN1-KO cells significantly reversed the increased clone formation observed upon chemotherapy drug treatment (Supplementary Fig. 7a, b), whereas Y333F BECN1 complementation in had no effect on the decreased clone formation mediated by chemotherapy drugs in these cells (Supplementary Fig. 7a, b). Subsequently, we performed an in vivo experiment to assess the

role of p-BECN1 (Y333) in CRC chemotherapy resistance in which subcutaneous tumor transplantation was performed in an immunocompetent mouse model of colorectal cancer. Because the amino acid sequence of the murine-derived BECN1 protein is different from that of human-derived BECN1, and the Y331 amino acid site of murine-derived Becn1 is identical to the Y333 amino acid site of human-derived BECN1, we first generated CT26-Becn1-KO, CT26-WT, CT26-Y331F, and CT26-Y331E cell lines. Then, these cells were subcutaneously injected into BALB/c mice before treatment with chemotherapy drugs. We observed that the tumor growth rate and tumor weight were significantly suppressed in the CT26-Becn1-KO groups (Supplementary Fig. 7c). Xenografts expressing WT Becn1 or Y331E Becn1 in CT26-Becn1-KO mice displayed a higher growth rate and a heavier weight (Fig. 6h, i). IHC results also demonstrated that caspase-3/PARP apoptotic signaling was activated in the CT26-Becn1-KO group but inhibited in WT or Y331E Becn1 complemented groups (Fig. 6j and Supplementary Fig. 7d). However, the Y331F Becn1 complemented groups showed high caspase-3/PARP apoptotic signaling pathway activation. Furthermore, LC3B expression was downregulated in the CT26-Becn1-KO group, while the inhibition of autophagy was rescued in the WT or Y331E Becn1

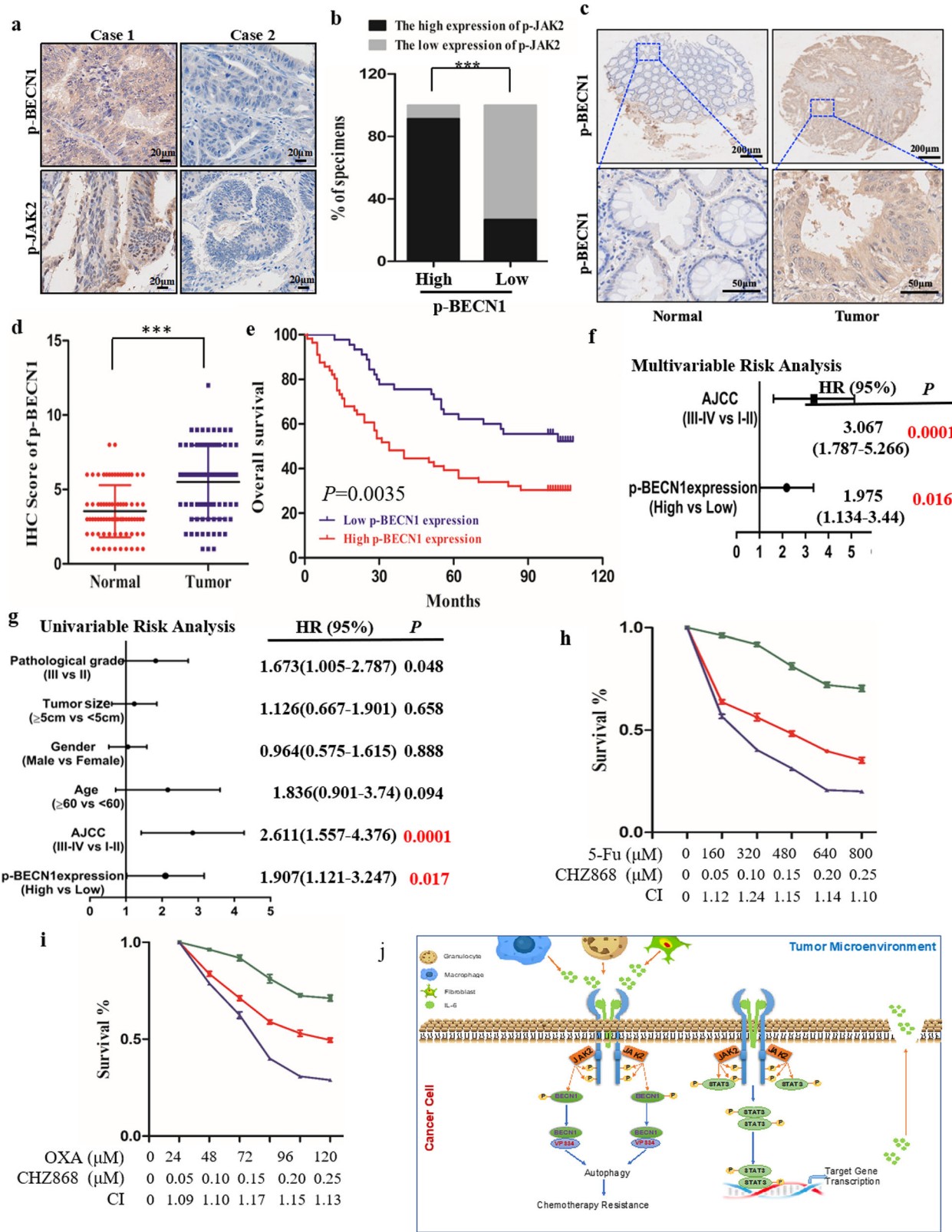

complemented groups, and Y331F Becn1 complementation only had a slight effect on autophagy (Fig. 6j and Supplementary Fig. 7d). Taken together, these data indicate that p-BECN1 (Y333) plays an important role in IL-6-induced CRC chemotherapy resistance.

**JAK2-induced BECN1 Y333 phosphorylation is a predictive marker of poor prognosis and chemotherapy resistance in cancer.** We subsequently investigated the clinical value of assessing JAK2-induced BECN1 Y333 phosphorylation in patients and observed that p-BECN1 (Y333) levels in CRC tissues (*Cohort-1*)

**Fig. 6 JAK2-induced BECN1 Y333 phosphorylation is a predictive marker for poor cancer prognosis and chemotherapy resistance. a** Representative results of immunohistochemical staining for p-JAK2 and p-BECN1 (Y333) in the same tumor tissues from 65 clinical CRC patients. Scale bars, 20 μm. **b** Statistical analysis of the expression of p-JAK2 and p-BECN1 (Y333) in tumor tissues. ***$P < 0.001$. The values are presented as the means ± s.d.; the p value ($\chi^2$-Test) is shown. **c** Representative results of immunohistochemical staining for p-BECN1 (Y333) in tumor tissues and adjacent normal tissues. Scale bars, 200 μm (top) and 50 μm (bottom). **d** Statistical analysis of p-BECN1 (Y333) levels in tumor tissues and adjacent normal tissues. ***$P < 0.001$. Data are presented as the means ± s.d., and the p value was determined by unpaired Student's t test. **e** Overall survival was compared between CRC patients with low and high levels of p-BECN1 (Y333) ($n = 44$ low p-BECN1 (Y333) levels; $n = 57$ high p-BECN1 (Y333) levels). Survival data were analyzed by the Kaplan–Meier method and log-rank test. **f**, **g** Univariate analysis (**f**) and multivariate analysis (**g**) were performed in *Cohort 2*. The bars correspond to 95% confidence intervals. **h**, **i** Synergy analysis of the JAK2 inhibitor CHZ868 and chemotherapy drugs (5-Fu or OXA) in LoVo cells. Cells were treated with the indicated concentrations of CHZ868 and 5-Fu (**h**) or OXA (**i**) for 36 h. The combination index (CI) value was examined. The CI value indicates the following: >1.15, synergism = 0.85–1.15, additive effect; and <0.85, antagonism. **j** Proposed schematic diagram of IL-6-mediated autophagy activation to promote chemotherapy resistance in colorectal cancer. Source data are provided in the Source Data file.

were positively correlated with p-JAK2 levels via IHC staining (Fig. 6a, b). Then, we examined the levels of p-BECN1 (Y333) between CRC samples and adjacent normal tissues in a CRC tissue microarray (*Cohort-2*). The IHC staining results were quantified as described in the Methods section, and we observed that high p-BECN1 (Y333) levels in CRC samples compared to normal tissues (Fig. 6c, d). We further assessed the relationship between the level of p-BECN1 (Y333) and the different clinicopathological characteristics in *Cohort-2*, and high p-BECN1 (Y333) levels were positively correlated with a poor CRC outcome (overall survival, OS) (Fig. 6e). Moreover, univariate and multivariate analyses indicated that the level of p-BECN1 (Y333) was an independent predictor of CRC outcome with significant hazard ratios in *Cohort-2* (Fig. 6f, g). These data demonstrate that p-BECN1 (Y333) is a potential predictor of CRC outcome. To further examine the clinical significance of the JAK2/BECN1 interaction in CRC, we investigated whether there is a synergistic effect between chemotherapy drugs and JAK2 inhibitors. Interestingly, we observed that the half maximal inhibitory concentration (IC50) of 5-Fu was significantly reduced by the JAK2 inhibitor CHZ868 (Fig. 6h). In addition, similar results were observed when OXA was used in combination with JAK2 inhibitors (Fig. 6i). Taken together, these results demonstrate that the JAK2/BECN1 complex plays an important role in CRC chemotherapy resistance and that p-BECN1 (Y333) is a potential predictor of CRC outcome and chemotherapy drug resistance.

## Discussion

Many cancer cell types induce multiple oncogenic signaling pathways to survive when the tumor microenvironment rapidly changes, such as under hypoxic conditions, in the presence of chemotherapeutic agents, and under nutrient deprivation[3]. In addition, many extracellular cytokines enriched in tumor tissues are involved in regulating various important properties of cancers to maintain cellular homeostasis, including autophagy, to adapt to outgrowth[4,34]. However, the precise molecular mechanisms underlying the link between autophagy and extracellular cytokines remain unknown. In the present study, we demonstrated that IL-6 activates autophagy in colorectal cancer, leading to the interaction between JAK2 and BECN1. Moreover, JAK2 functions as a protein kinase that phosphorylates BECN1 at Y333, to markedly promote the initiation of autophagy, which plays a crucial role in chemotherapeutic resistance.

Several lines of evidence suggest that cytokines enriched in the tumor microenvironment are deregulated in various cancers and contribute to a number of biological behaviors, including cancer proliferation, metastasis, and chemotherapeutic resistance[35]. In addition, many downstream protein kinases in oncogenic signaling pathways stimulated by extracellular cytokines, including EGFR[24], AKT[36], SIRT1[37], and AMPK[38], are considered key regulators of autophagy by directly or indirectly regulating

autophagy-related proteins (ATGs), suggesting that the initiation of autophagy regulated by extracellular cytokines may be dependent on multiple mechanisms. It has been well established that many ATGs undergo various posttranscriptional modifications (PTMs), including phosphorylation, acetylation, and methylation, which play important roles in the initiation of autophagy[39]. BECN1, a key autophagy regulator, is a crucial member of the class III phosphatidylinositol 3-kinase (PtdIns3K) complex. Furthermore, BECN1, which is modified by various protein kinases, is involved in the formation of autophagosome membranes for fusion with lysosomes. The posttranslational regulation of proteins is a major mechanism for the regulation of autophagy by BECN1. Previous studies have shown that the acetylation levels of BECN1 regulated by acetyltransferase p300 and deacetylase SIRT1 correlate with autophagy initiation[40]. Phosphorylation is considered a key modification of BECN1, and several reports have demonstrated that EGF/EGFR signaling negatively regulates autophagy by directly binding to BECN1 in cancer cells[24]. Interestingly, accumulating evidence indicates that BECN1 phosphorylation is directly involved in autophagy and tumor progression, including proliferation, metastasis, and tumor chemoresistance[24,41]. More importantly, the different sites of BECN1 phosphorylated by protein kinase are closely associated with downstream biological behavior[42]. These results are in line with previous consensus, suggesting that extracellular signals are involved in tumor survival partially through the regulation of autophagy [34]. Our observations indicate that JAK2, a nonreceptor tyrosine kinase, directly binds to BECN1 in the cytoplasm and modifies BECN1 Y333. We investigated the effect of JAK2-dependent phosphorylation of BECN1 on autophagy and observed that BECN1 Y333 phosphorylation strongly increased the initiation of autophagy, demonstrating that the phosphorylation of Y333 in BECN1 positively regulates the initiation of autophagy.

In recent years, many strategies targeting cell receptors in clinical tumor therapy have failed, although related extracellular cytokines have been shown to be upregulated in various types of tumors[43]. Our in vitro/vivo results are consistent with those of previous studies, showing that IL-6 overexpression leads to significant chemotherapy resistance in colorectal cancer. In addition, our data showed that IL-6 promotes the interaction between BECN1 and JAK2 and enhances BECN1 tyrosine phosphorylation at Y333, suggesting that the JAK2-dependent phosphorylation of BECN1 is the mechanism by which IL-6 promotes autophagy in colorectal cancer and underscoring the importance of autophagy promotion by IL-6/JAK2 signaling. Mechanistically, IL-6/JAK2 signaling-mediated phosphorylation of BECN1 at Y333 by contributes to chemotherapy resistance in colorectal cancer, and the pharmacological inhibition of autophagy leads to reestablishment of chemotherapy sensitivity. The initiation of autophagy induced by the IL-6/JAK2/BECN1 signaling pathway

appears to be a regulatory mechanism for chemotherapy resistance in colorectal cancer. Previous studies have shown that IL-6 augments cancer chemotherapeutic resistance through activation of the NF-κB κ and STAT3 signaling pathways[44,45]. However, the mechanisms underlying IL-6 leading to chemotherapy resistance in cancers have remains elusive. As a result, IL-6-targeting based therapies have limited benefits for tumor patients. Thus, combined with our present findings, both IL-6 and autophagy are potential targets for chemotherapy resistance in colorectal cancer.

Although targeted therapy has achieved good responses in the treatment of CRC, postoperative adjuvant chemotherapy and preoperative neoadjuvant chemotherapy are still the primary treatment options for advanced colorectal cancer. However, only ~40% of patients can benefit from chemotherapy, and chemotherapy resistance is the primary challenge faced by many colorectal cancer patients[46]. More regrettably, we cannot provide effective molecular targets to predict the sensitivity of patients to chemotherapy or identify chemotherapy resistance at an early stage. Tumors can only be used as chemotherapy resistance patient selection markers if they progress clinically. In the present study, we showed that BECN1 Y333 phosphorylation enhances BECN1 activation and promotes cancer chemotherapy resistance. Clinical sample data showed that the level of p-BECN1 (Y333) is related to CRC patient survival and is a predictive marker of a poor cancer prognosis. This finding may indicate that BECN1 Y333 phosphorylation can be used as a marker for the potential benefit of colorectal cancer chemotherapy and early identification of chemotherapy resistance.

Clinical investigation of the efficacy of JAK2 inhibitors in patients with solid tumors remains limited. Ruxolitinib, a JAK2-specific inhibitor, was shown to be well tolerated and potentially lead to improved survival outcomes in combination with capecitabine in a phase II study involving adults with metastatic pancreatic cancer. In the present study, we showed that JAK2 inhibitors combined with chemotherapy are more effective in treating CRC than chemotherapy alone. Although we provide limited data on cell line studies, this result may suggest that a phase II study of CRC patients with a combination of JAK2 inhibitor and chemotherapy could be worthwhile, especially in patients with BECN1 Y333 hypophosphorylation.

In summary, we elucidated a regulatory signaling pathway between extracellular cytokines and autophagy (IL-6/JAK2/BECN1 signaling pathway) and showed that JAK2-dependent BECN1 phosphorylation may contribute to chemotherapy resistance (Fig. 6j). Our data demonstrate that IL-6-induced autophagy contributes to the effectiveness of chemotherapy drug responses in colorectal cancer, and in combination with autophagy inhibitors or pharmacological agents targeting the IL-6/JAK2/BECN1 signaling pathway may represent a potential clinical therapy strategy. In addition, our results indicate that BECN1 Y333 phosphorylation can be used as a predictive marker for poor prognosis in CRC and as a marker for the potential benefit of colorectal cancer chemotherapy and the early identification of chemotherapy resistance.

## Methods

**Cell culture and reagents**. The human cancer cell lines HCT116, LoVo, MCF-7, and SW48; the prostate cancer cell lines PC3 and HEK293T; and CT26 murine colon cancer cells were purchased from the American Type Culture Collection and cultured in Dulbecco's modified Eagle's medium (DMEM, Cat# 12430054, Invitrogen, USA) supplemented with 10% fetal bovine serum (FBS, Cat# 10100147, Gibco, USA) and 100 U/mL penicillin and streptomycin. All cells were maintained at 37 °C in a 5% CO$_2$ incubator according to the needs of different experiments. Chloroquine (CQ; Cat#C6628) was obtained from Sigma. Inhibitors of STAT3 (Stattic, HY-13818) and JAK2 (CHZ868, Cat# HY-18960) were purchased from MedChemExpress. 5-Fluorouracil (5-Fu; Cat#F6627), oxaliplatin (OXA; Cat#O9512), IL-6 (Cat#SRP3096), and DMSO (Cat#D2650) were obtained from Sigma. The activity of the cells was examined with a Cell counting kit-8 (CCK8;

Cat#B34304, Bimake, USA). A PE Annexin V Apoptosis Detection Kit (Cat# 559763) was obtained from Becton Dickinson and Company. Lipofectamine 2000 (Cat# 11668019) was purchased from Invitrogen. The antibodies used in the present study were as follows: anti-MAP1LC3B (Cat#3868, Cell Signaling Technology, USA), anti-GAPDH (Cat#sc-32233, Santa Cruz Biotechnology, USA), anti-SQSTM1/p62 (Cat#66184-1-Ig, Proteintech Group, Wuhan), anti-VPS34 (Cat#12452-1-AP, Proteintech Group, Wuhan), anti-VPS15 (Cat# 17894-1-AP, Proteintech Group, Wuhan), anti-Bcl-2 (Cat#15071, Cell Signaling Technology, USA), anti-Rubicon (Cat#7151, Cell Signaling Technology, USA), anti-ATG14 (Cat#96752, Cell Signaling Technology, USA), anti-UVRAG (Cat#13115, Cell Signaling Technology, USA), anti-STAT3 (Cat#9139, Cell Signaling Technology, USA), anti-p-STAT3 (Y705) (Cat#9145, Cell Signaling Technology, USA), anti-GFP (Cat#2955, Qi dongzi Co., Wuhan), anti-JAK1 (Cat#29261, Cell Signaling Technology, USA), anti-p-JAK1 (Cat#74129, Cell Signaling Technology, USA), anti-JAK2 (Cat#3230, Cell Signaling Technology, USA), anti-p-JAK2 (Cat#ab32101, Abcam, USA), anti-BECN1 (Cat#3495, Cell Signaling Technology, USA and Cat#66665-1-Ig, Proteintech Group, Wuhan), anti-HA (Cat#3724, Cell Signaling Technology, USA), anti-Flag (Cat#14793, Qi dongzi Co., Wuhan), anti-P-Tyr-100 (Cat#9411, Cell Signaling Technology, USA), anti-caspase3 (Cat#9662, Cell Signaling Technology, USA), anti-cleaved caspase3 (Cat#9664, Cell Signaling Technology, USA), anti-Ki-67 (Cat#9449, Cell Signaling Technology, USA), anti-PARP (Cat#9532, Cell Signaling Technology, USA), anti-Cleaved PARP (Cat#5625, Cell Signaling Technology, USA) and anti-a-tublin (Cat#2148, Cell Signaling Technology, USA). A p-BECN1 (Y333)-specific antibody and the specific peptide for the BECN1 Y333 site were generated and purchased from Proteintech Group.

**Western blotting**. Cells were harvested and lysed in NP40 cell lysis buffer on ice for 30 min, after which the supernatants were collected by high-speed centrifugation. Then, the protein concentration of each sample was examined by using BCA protein reagent (Cat# 23252, Pierce, USA). Homogenate supernatants were resolved by SDS-PAGE, and the proteins were then transferred to PVDF membranes, after which the membranes were blocked in TBST solution supplemented with 5% milk. Primary antibodies were incubated with the appropriate membranes at 4 °C overnight. Then, the membranes were incubated with horseradish peroxidase-conjugated secondary antibodies at room temperature for 2 h. Finally, the membranes were detected with ECL reagents (Cat#35055, Thermo Scientific, USA) using Image Lab software.

**Plasmid construction, lentivirus packaging, and infection**. The plasmids encoding human JAK2 and BECN1 were cloned into pcDNA3.1(+). Site-directed mutagenesis of JAK2 (K882E) and BECN1 (WT, Y333F, and Y333E) was performed using a *Fast* Mutagenesis System Kit (Cat# FM111-01, TransGen Biotech, Beijing). All mutations were confirmed by DNA sequencing to avoid inaccurate mutations. The truncations of BECN1 (F1, F2, F3, and F4) and JAK2 (JH3-7, JH2, and JH1) were cloned into pcDNA3.1(+) by subcloning the wild-type plasmids of BECN1 and JAK2. The reconstructive plasmid vector of pLKOAS3W.puro, pMD2.G and psPAX2 were a gift from Hui-Kuan Lin lab. GFP-LC3B and BECN1 constructs (WT, Y333F, and Y333E) were cloned into pLKO AS3W.puro. Then, stable GFP-LC3B- and BECN1 (WT, Y333F, and Y333E)-expressing cells were constructed by lentiviral infection. Briefly, GFP-LC3B or BECN1 (WT, Y333F, and Y333E) plasmids, the psPAX2 packaging plasmid, and the pMD2.G envelope plasmid were mixed in a 2:1:1 ratio diluted in 250 μL serum-free DMEM and 15 μL Lipofectamine 2000. Then, the plasmid mixtures were transfected into HEK293T cells. The harvested viral supernatants were used to infect cancer cells. Stable cell lines were selected with 2 μg/ml puromycin for 2 days. The primers used in this study are shown in supplement Table 1. The plasmids were labeled with HA or Flag according to the needs of different experiments.

**Immunoprecipitation**. Cancer cells were harvested according to the needs of experiments. Then, the protein was extracted in NP40 cell lysis buffer supplemented with EDTA-free protease cocktail inhibitors for 30 min. The primary antibody was added to the supernatant of the cell lysate for rotation at 4 °C overnight. Then, magnetic beads (Cat#B26201 and Cat#B26101, Bimake, USA) or protein A/G agarose (Cat#sc-2003, Santa Cruz Biotechnology, USA) were added to cell lysates for incubation at room temperature. Subsequently, after 4 h, the beads or agarose were washed five times with NP40 cell lysis buffer supplemented with EDTA-free protease cocktail inhibitors. The immunoprecipitates were boiled in SDS-PAGE loading buffer and then used for western blot assays.

**Immunofluorescence**. For autophagosome and autophagic flux examination, cancer cells were plated on cover sides at 20% confluence in 24-well plates. After the treatment was performed as needed, the cells were washed with ice-cold PBS and fixed with 4% paraformaldehyde for 30 min at room temperature. Then, the cells were blocked in a 1% BSA solution in PBS for 1 h at room temperature and washed with PBS supplemented with 1% BSA three times followed by 0.1 mg/mL DAPI (Cat# 9542, Sigma, USA) for 10 min in the dark. Finally, the slides were washed with PBS and covered with glycerin. Cells were imaged using an Olympus FV-1000 confocal microscope. The number of LC3B puncta per cell in GFP-positive or mCherry-GFP-positive cells was determined as reported previously[47].

For colocalization immunofluorescence assays, cells were seeded on cover sides. After the treatment was performed as needed, the slides were fixed with 4% paraformaldehyde, blocked with PBS supplemented with 1% BSA and then permeabilized with 0.1% Triton-100. Next, slides were incubated with the primary antibody at 4 °C overnight. Then, the slides were embedded with fluorochrome-conjugated secondary antibody and incubated with DAPI in the dark for 10 min. Representative images were obtained using an Olympus FV-1000 confocal microscope.

**Colony formation assay**. One thousand cancer cells were seeded in six-well plates and incubated with DMEM supplemented with 10% FBS. After 24 h, chemotherapy drugs (5-Fu or OXA) were added to the cell medium. The cell medium was changed every 3 days. Cells were cultured under an atmosphere with 5% CO2 at 37 °C for 28 days, after which images of colonies were captured with a scanner.

**CCK-8 assay**. The viability of cancer cells was measured with a CCK-8 kit according to the manufacturer's instructions. Cancer cells were seeded in 96-well plates and pretreated with IL-6 (20 ng/mL) for 8 h. Then, the cells were cultured with chemotherapy drugs (5-Fu or OXA) alone or in combination with CQ (25 μl) for 36 h. Finally, 10 μl of CCK-8 reagent was added to the cell medium and incubated at 37 °C for 1 h. Cell viability was examined at 450 nm using a microplate reader.

**Flow cytometry**. Apoptosis was analyzed using flow cytometry according to the instructions of an Annexin V-PE/7-AAD Apoptosis Detection Kit. Cancer cells were seeded in 96-well plates and pretreated with IL-6 (20 ng/mL) for 8 h. Then, the cells were cultured with chemotherapy drugs (5-Fu or OXA) alone or in combination with CQ (25 μl) for 36 h. Subsequently, the cells were harvested in Annexin V binding buffer without EDTA and stained with Annexin V-PE (5 μl) and 7-ADD (5 μl) for 30 min in the dark at room temperature. Then, the cell apoptosis rate was measured by flow cytometry.

**Identification of BECN1-binding proteins and phosphorylation sites using public datasets**. The association between JAK2 and BECN1 was evaluated using online public datasets (https://string-db.org/), while potential residues in BECN1 phosphorylated by JAK2 were predicted using multiple public datasets (Kinase-Phos2: http://kinasephos2.mbc.nctu.edu.tw/and Kinexus|PhosphoNet: http://www.phosphonet.ca/) according to the online instructions.

**Molecular modeling of the interaction between BECN1-ECD and JAK2-JH1**. We modeled the complex structure between the ECD domain of BECN1 and the JH1 domain of JAK2 through protein–protein docking. The three-dimensional (3D) structures of the two proteins were downloaded from the Protein Data Bank (PDB), where the PDB ID for the JH1 domain is 3KRR and the PDB ID for the ECD domain is 4DDP. Then, MODELLER was used to add missing residues to the initial structure based on the sequence of the ECD domain. There were no missing residues in the initial structure of the JH1 domain except for the terminal residues. After adding the missing residues, the ECD domain structure was minimized using AMBER for 2000 steps. Then, we used the structure of protein kinase A (PKA) complexed with ATP (PDB ID: 1ATP) as the template to dock the ATP ligand into the JH1 domain by template-based docking. This ATP-bound JH1 structure was also minimized using AMBER for 20000 steps to remove the atomic clashes in the structure. Then, we used HDOCK to dock the refined BECN1-ECD domain structure against the ATP-bound JAK2-JH1 domain. HDOCK is a hybrid protein–protein docking algorithm that globally samples all possible binding poses with a fast Fourier transform (FFT)-based algorithm and evaluates the binding modes with ITScorePP[48]. HDOCK can also incorporate experimental information during the docking process. According to the literature, there are often two binding sites between kinases and substrates: the docking site and the catalytic site, where the ATP ligand is obviously in the catalytic site, and the docking region is typically formed by the G helix, part of the F helix and the region between them. Therefore, during docking procedure, we restrained residues 1048-1035 as the binding site residues in the JH1 domain. We also restrained the distance between residues 332-339 of the ECD domain and the ATP ligand to no more than 8 Å. After docking calculations, the docking results were clustered according to the docking scores with an RMSD cutoff of 5 Å and subjected to manual inspection for experimental information checks. Finally, the selected BECN1/ECD-JAK2/JH1 complex model was optimized using AMBER.

**Animal experiments**. To assess the role of IL-6 and BECN1 mutations in chemotherapy resistance, BALB/c female mice were bred in-house and subcutaneously injected with $1 \times 10^7$ tumor cells in 150 μl of PBS after obtaining approval by the Animal Experimentation Ethics Committee of Huazhong University of Science and Technology, Tongji Hospital.

To assess the role of IL-6 in colorectal cancer chemoresistance, three groups were evaluated: (1) DMSO, (2) IL-6, and (3) IL-6+CQ. After injection for 1 week, mice received an intratumoral injection of IL-6 (20 ng/g) alone or in combination with CQ (25 mg/kg) and intraperitoneal injection of oxaliplatin (5 mg/kg) or 5-FU (50 mg/kg) twice a week.

To assess the role of BECN1 mutation in colorectal cancer chemoresistance, five groups were evaluated: (1) CT26-WT-BECN1, (2) CT26-KO-BECN1 (Mock), (3) CT26-BECN-KO-WT, (4) CT26- BECN-KO-Y331F, and (5) CT26- BECN-KO-Y331E. Mice received an intraperitoneal injection of oxaliplatin (5 mg/kg) or 5-Fu (50 mg/kg) twice a week.

Tumor growth was assessed every other day by measuring tumor length (*L*) and width (*W*), and tumor growth was measured based on the formula (tumor volume = $(L \times W^2)/2$). After 3 weeks, the mice were killed, weighed, and imaged. IHC assays were performed to measure the expression of Ki-67, BECN1, LC3B, cleaved PARP and cleaved caspase 3. The tumor volume and weight results are shown as the means ± SEM ($n = 5$).

**Transmission electron microscopy**. Cells stably expressing GFP-LC3B were plated in 10-cm dishes. Then, after stimulation with IL-6 for 6 h, the cells were fixed in 2.5% glutaraldehyde for 12 h and then fixed in 1% osmium tetroxide for 1 h. Subsequently, samples were sent to the Wuhan Institute of Virology, where the remaining steps were performed. Finally, grids were subjected to transmission electron microscopy to observe mature autolysosomes. The autophagosomes were classified by size by determining the diameter (diameter > 500 nm), and the number of autophagosomes per cytoplasmic area was quantified by counting 3–5 cells per sample.

**Immunohistochemistry**. Clinical colorectal tumor tissue arrays (Cat# HCo-lA180Su15) were purchased from the Shanghai Outdo Biotechnology Co. Fresh tumor tissues from nude mice were fixed and embedded in paraffin, and IHC analyses were performed as previously described. Representative images were captured using a Nikon Eclipse Ni microsystem. The expression of related proteins was evaluated with an IRS system. Protein staining patterns were measured as low (IRS: 0-4) and high (IRS: 6-12). p-BECN1 staining intensity was categorized as follows: no staining, 0; weak, 1; moderate, 2; and strong, 3. The percentage of stained cells was categorized as follows: no positive cells, 0; <25% positive cells, 1; 25–50% positive cells, 2; 50–75% positive cells, 3; and >75% positive cells, 4. We calculated the score of each sample by multiplying the p-BECN1 staining intensity by the percentage of stained cells. For statistical analysis, scores of 0–4 were considered low expression, and scores of 6–12 were considered high expression.

**BECN1 knockout cell line construction**. BECN1 knockout (KO) cells were constructed using the CRISPR/Cas9 gene-editing system. The CRISPR plasmid was a gift from the Hui-Kuan Lin laboratory. Two gRNAs were designed to knockout BECN1 (5′-CACCgAAACTCgTgTCCAgTTTCAg-3′ and 5′-CACCgCCTg-gACCgTgTCACCATCC-3′). To select and confirm the knockout LoVo cell colonies, genomic DNA was extracted from both the parental and knockout LoVo cells. Then, PCR amplification was performed using primers binding ~1000 bp from the target site, and the PCR products were sequenced for validation. Western blotting was also performed to confirm the LoVo knockout cell lines. CT26-KO-Becn1 cells were constructed using a lentivirus containing the CRISPR/Cas9 gene-editing system to target the murine BECN1 gene, which was purchased from GeneChem Co. Then, CT26-KO-Becn1 cells were infected with lentivirus containing the BECN1 WT, BECN1 Y331F, or BECN1 Y331E genes to generate stable cell lines.

**Statistical analysis**. All data were analyzed with GraphPad Prism 8.0 (La Jolla, CA, USA) and are shown as the means ± SD or mean ± SEM. Where appropriate, the $\chi^2$-Test, two-tailed Student's $t$ test and Kaplan–Meier analysis were used. $P < 0.05$ was considered to indicate a significant difference.

**Reporting summary**. Further information on research design is available in the Nature Research Reporting Summary linked to this article.

## Data availability

Data for predicting the binding between JAK2 and BECN1 are accessible at String (https://string-db.org/). Data for the sites in BECN1 phosphorylated by JAK2 are available in multiple public datasets (KinasePhos2: http://kinasephos2.mbc.nctu.edu.tw/ and Kinexus|PhosphoNet: http://www.phosphonet.ca/). Data for the molecular modeling of the interaction between BECN1-ECD and JAK2-JH1 were obtained from PBD (https://www.rcsb.org/) and modeled with MODELLER (https://salilab.org/modeller/), HDOCK (http://hdock.phys.hust.edu.cn/) and AMBER (http://ambermd.org/). The data generated or analyzed during the current study are available within the article as well as the supplementary information file or from the corresponding author upon reasonable request. Source data are provided with this paper.

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

## Acknowledgements

We are grateful to Dr. Weina Zhang, Dr. Peijing Zhang, Dr. Min Wang, and Dr. Guoxiang Jin for the suggestions on this project. We thank Shengyou Huang and his lab members for the great favor of performing protein structure data analysis and project suggestions. We thank all the health workers for help against COVID-19 in Wuhan City and all coauthors for your great work during this difficult period. This work is supported by NSFC (No. 81773113 G.W., No. 81874186 J.H., and No. 81922053 G.W.) and startup funding from Tongji Hospital for G.W.

## Author contributions

FQ.H., J.H. and G.W. conceived and designed the experiments; FQ.H., C.H., C.S., D.S., Y.C., Q.W., AY.L., L.S., FY.H., and J.L. performed the experiments; Y.Y. and S.H. performed the protein structure analysis; F.X., Y.F., and X.L. provided suggestions for many experiments; and FQ.H., J.H., and G.W. organized and analyzed the data and wrote the manuscript.

## Competing interests

The authors declare no competing interests.
