## [Peer Review File · Nature Communications]

REVIEWER COMMENTS

Reviewer #1 (Remarks to the Author): with expertise in colorectal cancer and IL-6

Hu et al, "Tumor microenvironment derived IL-6 regulates autophagy and chemotherapy resistance by promoting BECN1 phosphorylation".

Authors use colon cancer cell lines to demonstrate how IL-6 whose levels are elevated during therapy and correlate with poor prognosis, induces autophagy in cancer cells and promotes resistance. They identify IL-6/JAK2/BECN1 pathway, particular interaction between IL-6 activated JAK2 and interaction between JAK2 and BECN1, which phosphorylates BECN1 at Y333. That phosphorylation is deemed essential for BECN1 activation and IL-6 induced autophagy by regulating PI3KC3 complex formation. BECN1 Y333 phosphorylation may serve as a predictive marker for a CRC poor prognosis and chemotherapy resistance. Overall, biochemical part of the paper which established IL-6-JAK2-Beclin1-autophagy connection is rather solid and novel. Human correlational data is useful. In vivo models provide some significant confirmation of the in vitro data however few points of criticism still apply, which are listed below.

Major:

- 1) Data on Fig1F seems to be a little bit counter-intuitive and needs quantification. It is not clear whether CQ actually induces autophagy (should not, right?) or inhibit it when used together with IL-6. Are these images representative and what is the "n" number for this experiment?
- 2) Why LC3B-II blots on Fig 1, 2 and 4 sometimes have 2 bands and sometimes 1 band? These blots and results should be presented consistently or these discrepancies explained?
- 3) Effects on Figure 4D is difficult to see and assess, and these should be as well quantified with the statistical analysis
- 4) It would be preferable to see the effects of beclin mutation, inhibition of autophagy and IL-6 in a immunocompetent mouse model in vivo using mouse colon cancer cell lines
- 5) Experiments on Fig 6I/h do not necessarily mean synergy as claims, as it could be easily an additive effect and the data presented is not enough to distinguish between these two possibilities

Minor:

- 1) Ref#1 is incomplete- no journal name- please check all of the references.

Reviewer #2 (Remarks to the Author): with expertise in autophagy

In the manuscripts, Hu et al reported the IL-6/JAK2/BECN1 pathway in the regulation of autophagy and chemotherapy resistant in colon cancer. Particularly, in contrast to previous findings, the IL-6-JAK2-Beclin1 Y333 phosphorylation axis is shown to be essential for IL-6-stimulated autophagy, which also plays roles in tumor resistance to chemotherapy. Most experiments are logically designed and the new phosphorylation of Beclin1 Y333 by IL-6 signaling represents a novel mechanism of autophagy regulation, particularly in the context of tumor-tumor microenvironment interaction. However, experiments for the implication of this new mechanism in tumor resistance and clinical relevance are largely flawed and need to be seriously re-designed and validated.

Major Points

- Autophagy assay throughout the manuscript needs to be improved, as detailed below:
- it is difficult to tell autophagosomes difference between the untreated and IL6-treated groups in electron microscopy data in Fig. 1d-e. Increased resolution is needed.
- Wrong data presentation in Fig. 1f and S1e. If IL-6 could promote autophagic flux as noted by the authors, adding CQ could only further increase, rather than decrease, autophagosome

accumulation by blocking lysosomal degradation. However, Fig. 1e showed that the overall autophagy levels dropped upon IL-6+CQ treatment in comparison with that of IL-6 or CQ group. This experiment needs to be repeated. Fig. 1f images are not convincing either. Quantifying % of cells with LC3 puncta is not a reliable way to reflect autophagic flux in cells in the experimental setting of this work. It is hard to understand IL-6 counteracts CQ effects, and if so, additional mechanisms might exist for this regulation.

- Why increased LC3B-II is accompanied by increased rather than decreased LC3B-I in S1a and S1b. Does IL-6 induce LC3 gene expression?
- LC3 blot in Fig. S1g is inconsistent with most other LC3 blots.
- IL-6/JAK is known to activate the PI3K/Akt/mTOR pathway, which negatively regulates autophagy. How this signaling even impacts on the IL-6/JAK-mediated autophagy activation?
- Fig. 2a p62 blot failed to reflect IL-6-mediated autophagy induction

- JAK2 independence

- The reduction of LC3 puncta in IL-6+CHZ treated SW48 cells in Fig. 2b were not clear and did not match the immunoblot analysis of LC3B in Fig. 2a
- Please provide western blot analysis of JAK1 for all JAK2-related data to assure that the inhibitor and/or siRNA against JAK2 did not have off-target effect particularly to the closely related JAK1
- Why overexpressing JAK2 decreases the expression of BECN1 in Figure S3b, while BECN1 overexpression decreases the expression of JAK2 in Figure S3c?
- JAK2 blot is missing in Fig. 3c and S3e, which is an important control and should be included
- The domain mapping data only indicates that the JH-1 domain of JAK2 and the ECD of Beclin1 are sufficient for binding, but not necessarily indispensable.
- Since both Vps34 and Jak2 target the same ECD domain of Beclin1, do they compete for Beclin1 interaction?

- Y333 phosphorylation

- Since the authors noted the existence PI3KC3-C1 and -C2 in autophagy regulation, to evaluate the effect of Y333 phosphorylation on IL-6-induced Beclin1 interactome, more in-depth analyses is recommended. Only evaluating Beclin1 interaction with Vps34 or Vps15 is limited and won't reflect interactome regulation; other complex subunits or regulators such as UVRAG, Atg14, Rubicon, and Bcl-2 (inhibitor of Beclin1) should also be included in Fig. 4a-b. Also the difference between Y333E vs 333F on Beclin1 interaction with Vps34 is not evident in Fig. 4c and that data needs to be repeated.
- Beclin1 interaction with Vps34 and Vps15 is quite dominant and easily detected even under physiological level, however, it is hardly seen in Fig.4a.

- Chemotherapy resistance

- It is not surprising at all to see IL-6 treatment reduced cell death induced by chemotherapy reagents given that IL-6 turns on a series of cell proliferation and anti-apoptosis pathway. To validate this increased chemoresistance is due to Y333-phosphorylated Beclin1 and resultant autophagy but not the other well established anti-apoptosis pathway, the authors may want to test cell viability of 333F cells in the presence of IL-6 and chemo-drug. CQ is not a validated drug combination in this context because recent studies have shown many autophagy-independent anti-tumor mechanism of CQ, thus even though IL6+CQ showed less resistance, it could not corroborate the new mechanism identified in this study.
- Unless Beclin1 Y333 phosphorylation is constitutively induced, Fig. S7a should include IL-6 treatment for colony formation to verify the importance of IL-6-induced Y333 phosphorylation in 2D cell growth. At least according to the data of Fig. 4d and e, no much difference in basal autophagy was observed between WT and Y333F, suggesting this modulation is not fully functioning in basal level. This is not in agreement with the significant colony formation between these mutants in S7.
- Fig. S7 should be re-conducted in the presence of IL-6 to make the mechanism identified in the right context.
- For chemoresistance, tumor growth curve should be provided rather than the endpoint data of tumor weight.
- Xenograft animal experiments lack details in method. There is no clue how many cells were injected and what is the scheme to apply treatment.
- It is surprising to see no difference in tumor growth between different groups under non-treated

condition, which is against most published work and need to be further verified.

- In the absence of IL-6, why do we expect to see such significant difference in tumor weight between WT and Y333F group in Fig. 5h and S7c, given that their basal level of autophagy seemed comparable (refer to Fig. 4d-e). This could be the case if IL-6 is not the unique mechanism to drive 333 phosphorylation and some other factors may do.
- Autophagy levels in the dissected tumors should be evaluated to confirm the identified role of Y333 in autophagy regulation in xenograft tumor in vivo.

- Clinical relevance
- Unclear how the high and low p-BECN1 is defined.
- It remains unclear why Y333-p is associated with significant hazard ratios given that both cell-based assay and xenograft did not show much difference in tumor progression between WT and mutants
- Combination of JAK2 inhibitor and chemo-drugs, though showing significant effect on cells survival, did not bring further insight into the Y333 mechanisms. Note the downstream umbrella like effect of JAK2 on cell proliferation, differentiation, and apoptosis.

Minor Points

- Please provide the link of the multi-database analysis for Tyr identification in the legend of Fig. S4c.
- Fig. 3e, there seems to be more JAK2 WT exogenous expression in WCE than that of K882E mutant; Fig 3F the reduced Tyr phosphorylation of Beclin1 upon JAK2 K882E expression seems not as clear as expected
- Fig. 4a and 4b could be merged into one graph so that both the basal levels of complex formation and IL-6-mediated increased assembly could be compared in parallel with JAK2 inhibitor.
- Fig S6b PARP1 blots need to be repeated. The un-cleaved PARP and cas-3 should be included as controls
- Fig.6c the marginal region of Normal tissue also showed position staining of p-BECN1, casting doubt on the antigen specificity of this antibody.
- Do high levels of p-BECN1 in patient samples correlate with IL-6 too?

Reviewer #3 (Remarks to the Author): with expertise in autophagy

Interleukin-6 is tightly tied to tumorigenesis and multiple reports have shown that IL-6 stimulates autophagy in cell culture settings. The mechanism of autophagy activation is however debated. Transcriptional activation of autophagy is rare and evidence for STAT3-mediated control of autophagy is weak.

This manuscript claims that IL-6 activates autophagy not through transcriptional responses mediated by STAT3, but rather through upstream activating Tyrosine phosphorylation by JAK on BECLIN1, a component of a major autophagy-regulating lipid kinase complex. This conveys resistance to anticancer treatment by 5-FU and OXA in vitro. Correlative evidence from colorectal cancer patient samples indicate that phospho-JAK and phospho-Becclin1 status predict poorer survival.

The authors first recapitulate earlier results that IL-6 induces autophagy and find that this does depend on Janus Kinase 2 (JAK2) but not the major downstream signalling transducer, STAT3 by knock-out/knock-down and pharmacological means. They move on to show that JAK2 can physically interact with BECN1. Candidate putative direct phosphorylation sites on BECN1 by JAK2 is modelled and tested and Y333 phosphorylation is stimulated by JAK2 activity in vitro and affects the autophagy-regulating VPS34 lipid kinase assembly suggesting a likely direct mechanism for autophagy stimulation. IL-6 mediated resistance to chemotherapy induced cell death is known to depend on autophagy. It is demonstrated that it also depends on the JAK mediated BECN1Y333 phosphorylation in vitro and in a mouse xenograft model. Some correlative patient data suggests that Y333 phospho-BECN1 predicts poor survival suggesting that the JAK2-BECN1-Autophagy axis is relevant to human cancer.

The identification of a direct regulation of autophagy through BECLIN1 by JAK is novel and gives valuable insight in mechanistic control of autophagy of relevance to tumorigenesis. It will be of considerable interest to a wide readership in cell biology, cell signalling, autophagy, immunology and cancer biology.

The manuscript is technically sound and convincing if taken at face value, but source data and statistics, in particular for most Western Blots, are lacking and must be provided to . Co-localization data based on immune fluorescence microscopy is not currently convincing.

The title of the paper "Tumor microenvironment derived IL-6 regulates autophagy and chemotherapy resistance by promoting BECN1 phosphorylation" is not entirely justified as the functional and mechanistic studies are carried out mostly in vitro and the source of IL-6 not addressed. Removing "Tumor microenvironment derived" from the title is an option.

Numerous statements are inaccurate due to poor English and need to be corrected.

Comments:

1. Using two different cell lines (LoVo and SW48) autophagy flux upon IL-6 stimulation is evaluated using well established autophagy markers: Autophagic GFP-LC3 puncta formation by microscopy, LC3-II (lipidated form) indicating autophagy activation and decrease of the autophagy substrate SQSTM1, and lysosomal release of "free" GFP from GFP-LC3.

On the surface, the experiments are sound and carefully performed.

However, throughout the paper all Western Blots lack molecular weight markers, number of repeats performed and quantifications + statistics of intensities. The number of independent experiments, source data (Excel files and whole Western blots, not cropped) must be provided. It is impossible to evaluate, with the current images provided, whether correct identification of autophagosomes by transmission electron microscopy has been performed. Please provide better images (higher magnification) in Fig1d and/or supplementary material. I could not find the description of how this quantification was performed in the materials and methods (methods chosen, randomizations, repeats, # of cells, cytoplasmic area covered).

2. In Fig1-4 please signify the expected size of LC3 I and II with molecular weight marker/size indicated on the blots.

3. Evaluation of co-localization of JAK and BECN1 by immune fluorescence is not possible at this magnification. Please provide high magnification images using a 63X or 100X objective or even better, super resolution imaging. Provide scale bars.

4. In my opinion, too much stress and space is used at describing the molecular modelling and in Fig3h. It is a useful addition, but based on minimal amounts of experimental data and caution should be exercised to avoid overreach. Statements like "The data shows the modelled structure,...." should not be used.

5. Figure 4. Autophagy induction by IL-6 is not evident without quantifications of LC3II vs LC3I, disappearance of SQSTM1 upon stimulation. Provide repeats and quantifications and statistics. Source data, repeats, number of cells, dots and experimental details are lacking for 4g. unless this is provided in files I could not open, this is symptomatic. Please make sure that all data are backed with sufficient experimental detail and source data.

6. The term "inactive" unphosphorylated mutants for BECN1 Y333F and Y338F). It is more accurate to use phosphorylation defective. Similarly, the term "active" BECN1 is used for Y333E. As BECN1 is not an enzyme and it is unclear what active/activity means in this setting, the authors are advised to stick to "phospho mimetic".

Minor comments:

Introduction:

«Autophagy, a highly conservative biological behavior that is a lysosomal-dependent degradation process, plays a vital role in cellular protection by recycling degradation «

This is inaccurate in several ways: rewrite

Response to the comments of the reviewers (NCOMMS-20-29331-T)

We would like to thank all the reviewers for recognizing the innovation and importance of our study and for providing constructive comments that in turn help strengthen our study tremendously. We have provided point-by-point answers to address all the reviewers' comments. Importantly, we have provided compelling new experimental data to address the reviewers' concerns.

The point-by-point responses to the reviewers' comments are as followed:

Reviewer #1:

Hu et al, "Tumor microenvironment derived IL-6 regulates autophagy and chemotherapy resistance by promoting BECN1 phosphorylation". Authors use colon cancer cell lines to demonstrate how IL-6 whose levels are elevated during therapy and correlate with poor prognosis, induces autophagy in cancer cells and promotes resistance. They identify IL-6/JAK2/BECN1 pathway, particular interaction between IL-6 activated JAK2 and interaction between JAK2 and BECN1, which phosphorylates BECN1 at Y333. That phosphorylation is deemed essential for BECN1 activation and IL-6 induced autophagy by regulating PI3KC3 complex formation. BECN1 Y333 phosphorylation may serve as a predictive marker for a CRC poor prognosis and chemotherapy resistance. Overall, biochemical part of the paper which established IL-6-JAK2-Beclin1-autophagy connection is rather solid and novel. Human correlational data is useful. In vivo models provide some significant confirmation of the in vitro data.

Response: We thank the reviewer for recognizing the novelty and importance of our study. We also thank the reviewer for providing the insightful and constructive comments that help strengthen our conclusion further.

Major Points:

1) Data on Fig1F seems to be a little bit counter-intuitive and needs quantification. It is not clear whether CQ actually induces autophagy (should not, right?) or inhibit it when used together with IL-6. Are these images representative and what is the "n" number for this experiment?

Response: We thanks for reviewer pointing out this question. As reported in our manuscript and in previous literature, autophagy is a highly dynamic, multi-step process, which begins with sequential formation of autophagosomes and autolysosomes, where the engulfed constituents are degraded. Thus, to assess the autophagy activity mediated by IL-6, we used the tandem monomeric mCherry-GFP-tagged LC3. The GFP signal is sensitive to the acidic condition of the lysosome lumen, whereas mCherry is more stable. Therefore, colocalization of both GFP and mRFP fluorescence indicates a compartment that has not fused with a lysosome, such as an autophagosome. In contrast, a mCherry signal without GFP

corresponds to an autolysosome. In addition, chloroquine (CQ) treatment caused the accumulation of autophagosomes by blocking fusion with the lysosome to inhibit autophagy flux. Under autophagy flux was inhibited, the detection of autophagosome reflected autophagy activity. This is the accepted method to detect autophagy activity. Thus, to assess the autophagy activity mediated by IL-6, CQ was added into the culture medium to inhibit the formation of autolysosomes (red signal). As shown in Revised Figure.1f, CQ treatment actually decreased the red signal, meanwhile, IL-6 treatment increased the red signal while the presence of CQ. The date of quantification of Figure 1f was shown in supplementary Figures as Figure.S1e. In order to make these images more representative, we repeated this part of the experiment for **three times**. Original results are also provided as **Revised Figure.S8**.

2) *Why LC3B-II blots on Fig 1, 2 and 4 sometimes have 2 bands and sometimes 1 band? These blots and results should be presented consistently, or these discrepancies explained?*

Response: Thanks for reviewer's carefully reading and professional commenting our manuscript. This difference of LC3B-II blots was caused by different exposure time. To better explain the different phenomenon, we increased the exposure time of these original bands and then we found all in this manuscript LC3B blots have 2 bands. We also provided the original bands in raw data as **Revised Figure.S8**.

3) *Effects on Figure 4D is difficult to see and assess, and these should be as well quantified with the statistical analysis*

Response: Thanks for reviewer's suggestion, we have added the quantification data into **Revised Figure.4d**.

4) *It would be preferable to see the effects of beclin mutation, inhibition of autophagy and IL-6 in an immunocompetent mouse model in vivo using mouse colon cancer cell lines.*

Response: Thank you for reviewer's suggestion. We repeated animal experiments using mouse colon cancer cell lines (CT26). As shown in **Revised Figure.5e, f**, inhibition of autophagy strongly decreased the chemoresistance mediated by IL-6. Due to the amino acid sequence of the murine-derived Becn1 protein is different to the human-derived BECN1 and the Y331 amino acid site of the murine-derived Becn1 is identical to the Y333 amino acid site of the human-derived BECN1, we constructed Beclin1 knockout CT26 cell lines, WT-Beclin1, Y331F-Beclin1, Y331E-Beclin1 CT26 cell lines. Then, cells were subcutaneously injected into immunocompetent mouse to examine the relationship between p-Y331 Beclin1 and tumor chemoresistance. As shown in **Revised Figure.5h, i**, we showed that tumor growth rate and tumor weight were significantly suppressed in CT26-Beclin1-KO groups and CT26-Y331F Beclin1 groups. WT Beclin1 or Y331E Beclin1 groups mice displayed a higher growth rate and a heavier weight of xenografts. Detailed results were also been described in the revised manuscript. Taken together, we confirmed that the effects of Beclin1 mutation, inhibition of autophagy and IL-6 in an

immunocompetent mouse model in vivo using mouse colon cancer cell lines is consistent with the data showed in human colon cancer cell lines.

5) *Experiments on Fig 6I/h do not necessarily mean synergy as claims, as it could be easily an additive effect and the data presented is not enough to distinguish between these two possibilities.*

Response: Thank you for reviewer pointed out this important issue. The coefficient of drug interaction (CI) was usually used to analyze the synergistically inhibitory effect of drug combinations, this approach also had been used in other studies[1-3]. There are several methods to calculate CI. In this manuscript, we used Bürgi formula to measure the coefficient of drug interaction (CI). The value of CI =1 would mean simple addition; >1 means synergism or potentiation and <1 means antagonism. Thus, we used CI here to present the synergy effect of treatment with two drugs.

- [1] Karakashev S, Fukumoto T, Zhao B, Lin J, Wu S, Fatkhutdinov N, Park PH, Semenova G, Jean S, Cadungog MG, Borowsky ME, Kossenkov AV, Liu Q, *et al.* EZH2 Inhibition Sensitizes CARM1-High, Homologous Recombination Proficient Ovarian Cancers to PARP Inhibition. *Cancer Cell* 2020; 37: 157-167.e156.
- [2] Parasramka M, Yan IK, Wang X, Nguyen P, Matsuda A, Maji S, Foye C, Asmann Y, Patel T. BAP1 dependent expression of long non-coding RNA NEAT-1 contributes to sensitivity to gemcitabine in cholangiocarcinoma. *Mol Cancer* 2017; 16: 22.
- [3] Hou Z, Sun L, Xu F, Hu F, Lan J, Song D, Feng Y, Wang J, Luo X, Hu J, Wang G. Blocking histone methyltransferase SETDB1 inhibits tumorigenesis and enhances cetuximab sensitivity in colorectal cancer. *Cancer Lett* 2020; 487: 63-73.

Minor Points:

1) *Ref#1 is incomplete- no journal name- please check all of the references.*

Response: Thank you for your suggestions. We are sorry for our mistakes. We have corrected in revised manuscript and carefully checked all of the reference.

Reviewer #2

In the manuscripts, Hu et al reported the IL-6/JAK2/BECN1 pathway in the regulation of autophagy and chemotherapy resistant in colon cancer. Particularly, in contrast to previous findings, the IL-6-JAK2-Beclin1 Y333 phosphorylation axis is shown to be essential for IL-6-stimulated autophagy, which also plays roles in tumor resistance to chemotherapy. Most experiments are logically designed and the new phosphorylation of Beclin1 Y333 by IL-6 signaling represents a novel mechanism of autophagy regulation, particularly in the context of tumor-tumor microenvironment interaction. However, experiments for the implication of this new mechanism in tumor resistance and clinical relevance are largely flawed and need to be seriously re-designed and validated.

Response: We thank the reviewer for providing the insightful and constructive comments. We take reviewer's comments seriously and have provided compelling new experimental data to address the reviewers' concerns.

Major Points:

Autophagy assay throughout the manuscript needs to be improved, as detailed below: it is difficult to tell autophagosomes difference between the untreated and IL6-treated groups in electron microscopy data in Fig. 1d-e. Increased resolution is needed.

Response: Thank you for your suggestions. We have repeated the data of electron microscopy and improved the picture resolution as your required. We hope that the revised images are acceptable.

Wrong data presentation in Fig. 1f and S1e. If IL-6 could promote autophagic flux as noted by the authors, adding CQ could only further increase, rather than decrease, autophagosome accumulation by blocking lysosomal degradation. However, Fig. 1e showed that the overall autophagy levels dropped upon IL-6+CQ treatment in comparison with that of IL-6 or CQ group. This experiment needs to be repeated.

Fig. 1f images are not convincing either. Quantifying % of cells with LC3 puncta is not a reliable way to reflect autophagic flux in cells in the experimental setting of this work. It is hard to understand IL-6 counteracts CQ effects, and if so, additional mechanisms might exist for this regulation.

Response: Thank you for this important issue and giving challenge comments. We re-examined the data showed in Figure.1f and S1e and repeated this experiment. As you pointed out, adding CQ actually increased autophagosome accumulation. We have corrected it in revised manuscript, and we are so sorry for this mistake. Meanwhile, chloroquine (CQ) treatment caused the accumulation of autophagosomes by blocking fusion with the lysosome to inhibit autophagy flux. Following autophagy flux was inhibited, the detection of autophagosome may predict autophagy activity. We repeat the experiments and provided the original data in our revised manuscript and figures.

Why increased LC3B-II is accompanied by increased rather than decreased LC3B-I in S1a and S1b. Does IL-6 induce LC3 gene expression?

Response: Thanks for reviewer's question. We repeat the experiments and give the consistent data in **Revised Supplementary Figure.S1a and S1b**.

LC3 blot in Fig. S1g is inconsistent with most other LC3 blots.

Response: Thank you for carefully reading and professional commenting our manuscript. This difference of LC3B-II blots was caused by different exposure time. To better explain the different phenomenon, we increased the exposure time of these original bands and then we found all in this manuscript LC3B blots have 2 bands. We also provided the original bands in raw data as **Revised Figure.S8**.

IL-6/JAK is known to activate the PI3K/Akt/mTOR pathway, which negatively regulates autophagy. How this signaling even impacts on the IL-6/JAK-mediated autophagy activation?

Response: Thank you for putting forward the important suggestion. As you pointed, other studies showed that IL-6/JAK could activate the PI3K/AKT/mTOR pathway to inhibit autophagy, in addition, there are other indirect signaling networks of IL-6 downstream that also regulate autophagy. In our findings, we proved that the autophagy promotion effect of IL-6 is dependent on the phosphorylation of BECN1 and focused on elucidate the direct signaling pathway of IL-6 regulating autophagy in cancer cells. In this study, we did not have many data to show that the PI3K/Akt/mTOR pathway regulates IL-6/JAK-mediated autophagy activation, but it is an interesting question and maybe we can study in our future studies.

Fig. 2a p62 blot failed to reflect IL-6-mediated autophagy induction.

Response: The results of Fig.2a showed that IL-6 inhibited the expression of p62. We gave the quantified data in **Revised Figure.2a** and give our original data as **Revised Figure.S8**.

JAK2 independence

The reduction of LC3 puncta in IL-6+CHZ treated SW48 cells in Fig. 2b were not clear and did not match the immunoblot analysis of LC3B in Fig. 2a.

Response: Thank you for reviewer's question. To improve the quality of this data, we repeated this experiment and data were presented in **Revised Figure.2b**.

Please provide western blot analysis of JAK1 for all JAK2-related data to assure that the inhibitor and/or siRNA again JAK2 did not have off-target effect particularly to the closely related JAK1

Response: Thank you for your suggestions. We have repeated related experiment and provided western blot results of JAK1 according your suggestions. As shown in **Revised Figure.2e** and **Revised Figure.4a**, the inhibitor and siRNA again JAK2 is specific. These results were added into the revised manuscript, which excluded off-target effect to the JAK1.

Why overexpressing JAK2 decreases the expression of BECN1 in Figure S3b, while BECN1 overexpression decreases the expression of JAK2 in Figure S3C?

Response: In Figure.S3b experiment, we tried to prove the interaction between BECN1 and JAK2, we transfected both Flag-JAK2 and HA-BECN1 plasmids into 293T cells, the co-transfection may affect two plasmids transfection efficiency. In addition, we use the anti-HA and anti-Flag for this western blot study, the results can not reflect the endogenous protein expression level and the regulation relationship between JAK2 and BECN1.

JAK2 blot is missing in Fig. 3c and S3e, which is an important control and should be included.

Response: Thanks for reviewer's suggestion. We have repeated the experiments and examined JAK2 expression. The results have added into the **Revised Figure.3c and S3e**.

The domain mapping data only indicates that the JH-1 domain of JAK2 and the ECD of Beclin1 are sufficient for binding, but not necessarily indispensable.

Response: The domain mapping data were explained the possible joint amino acid sequence position between JAK2 and BECN1. We have modified the statement of this section to make the statement more precise and rigorous.

Since both Vps34 and Jak2 target the same ECD domain of Beclin1, do they compete for Beclin1 interaction?

Response: Thank you for your question. In Figure 4a, we observed that IL-6 promoted the binding of BECN1 and VPS34, while inhibition of JAK2 activity inhibited the binding of BECN1 and VPS34. If Vps34 and JAK2 compete for BECN1, the interaction between BECN1 and VPS 34 should be increased after JAK2 activity was inhibited. However, we didn't find this phenomenon.

Y333 phosphorylation

Since the authors noted the existence PI3KC3-C1 and -C2 in autophagy regulation, to evaluate the effect of Y333 phosphorylation on IL-6-induced Beclin1 interactome, more in-depth analyses are recommended. Only evaluating Beclin1 interaction with Vps34 or Vps15 is limited and won't reflect interactome regulation; other complex subunits or regulators such as UVRAG, Atg14, Rubicon, and Bcl-2 (inhibitor of Beclin1) should also be included in Fig. 4a-b. Also the difference between Y333E vs 333F on Beclin1 interaction with Vps34 is not evident in Fig. 4c and that data needs to be repeated.

Response: Thank you for your suggestions. We have repeated those experiments and meanwhile evaluated the relationship between BECN1 and other regulators such as UVRAG, Atg14, Rubicon, and Bcl-2. We added related results in **Revised Figure.4a, b**.

Beclin1 interaction with Vps34 and Vps15 is quite dominant and easily detected even under physiological level, however, it is hardly seen in Fig.4a.

Response: Thank you for your reminding. This binding difference is likely due to cell type. Of note, differences in antibody manufacturers and exposure times also may result in weaker bands. To observe more brighter bands, we optimized the experimental conditions and enhanced the exposure time. The results of repeated experiments are added in the **Revised Figure.4a, b**.

Chemotherapy resistance

It is not surprising at all to see IL-6 treatment reduced cell death induced by chemotherapy reagents given that IL-6 turns on a series of cell proliferation and anti-apoptosis pathway. To validate this increased chemoresistance is due to

Y333-phosphorylated Beclin1 and resultant autophagy but not the other well established anti-apoptosis pathway, the authors may want to test cell viability of 333F cells in the presence of IL-6 and chemo-drug. CQ is not a validated drug combination in this context because recent studies have shown many autophagy-independent anti-tumor mechanisms of CQ, thus even though IL6+CQ showed less resistance, it could not corroborate the new mechanism identified in this study.

Response: Thank you for your challenge question. At present, there is no specific autophagy inhibitor. From the clinical point of view, most of the current studies also use chloroquine as an autophagy inhibitor for research. In this manuscript, we mainly elucidated the effect of IL-6 on autophagy by the regulation of BECN1 phosphorylation. Further, we constructed activated and inactivated plasmids of BECN1Y333 and demonstrated the role of BECN1Y333 on autophagy and chemotherapy resistance. These data are sufficient to demonstrate that IL-6 may indeed regulate autophagy by mediating BECN1 phosphorylation, thereby affecting chemotherapy-resistance.

Unless Beclin1 Y333 phosphorylation is constitutively induced, Fig. S7a should include IL-6 treatment for colony formation to verify the importance of IL-6-induced Y333 phosphorylation in 2D cell growth. At least according to the data of Fig. 4d and e, no much difference in basal autophagy was observed between WT and Y333F, suggesting this modulation is not fully functioning in basal level. This is not in agreement with the significant colony formation between these mutants in S7.

Response: In Figure.S7, we actually constructed activated and inactivated plasmids of BECN1Y333, which simulated the conditions treated with or without IL-6. Therefore, this part of the experiment does not need to re-add IL-6. In addition, as you pointed, according to the data of Figure. 4d and Figure.4e, not much difference in basal autophagy was observed between WT and Y333F, suggesting this modulation is not fully functioning in basal level. In Figure.S7, colony formation between those mutants was no significant difference under normal conditions, while significant difference between those mutants was showed under cell treated chemotherapy drugs. As we known, autophagy in tumors usually functions as a protective mechanism when cells are subjected to external stress. Thus, the significant difference of colony formation between these mutants was found under cell treated with chemotherapy drugs, but not without chemotherapy drugs.

Fig. S7 should be re-conducted in the presence of IL-6 to make the mechanism identified in the right context.

Response: In this manuscript, we mainly elucidated that IL-6 regulates autophagy by mediating phosphorylation of Y333, thus affecting chemotherapy resistance of tumors. In Figure. S7, we constructed activated and inactivated plasmids of BECN1Y333, which simulated the conditions treated with or without IL-6. Therefore, this part of the experiment does not need to re-add IL-6.

For chemoresistance, tumor growth curve should be provided rather than the endpoint data of tumor weight.

Response: Thank you for professional comments. The data of tumor growth curve was provided in **Revised Figure.5i, f, g** in primary manuscript.

Xenograft animal experiments lack details in method. There is no clue how many cells were injected and what is the scheme to apply treatment.

Response: Thanks for your reminding. We carefully provided the details of xenograft animal experiments in the section of revised method and the related data was added into the revised manuscript.

It is surprising to see no difference in tumor growth between different groups under non-treated condition, which is against most published work and need to be further verified.

Response: In xenograft animal experiments, tumor growth curve data revealed that the tumor growth curve of IL-6 treat groups was steeper than that of the other two group under non-treated condition. Of note, with the extension of the experimental time, the difference of the tumor growth among three groups would be significant. Since the purpose of our study was to observe the role of autophagy in tumor chemotherapeutic resistance, there were a few transplanted tumor cells injected, and tumor growth was kept in a state of adequate nutrition as far as possible, excluding the influence of other factors on autophagy and chemotherapy drug. Thus, the differences between the groups were not significant. It wasn't conflict with previous documents reported.

In the absence of IL-6, why do we expect to see such significant difference in tumor weight between WT and Y333F group in Fig. 5h and S7c, given that their basal level of autophagy seemed comparable (refer to Fig. 4d-e). This could be the case if IL-6 is not the unique mechanism to drive 333 phosphorylation and some other factors may do.

Response: This is an interesting phenomenon. We hypothesis it is related to the role of autophagy in different tumor environments. We know that autophagy in tumors usually functions as a protective mechanism when cells are subjected to external stress and under normal conditions autophagy plays slight roles in tumor progression. In WT and 333F groups, the expression of BECN1 phosphorylation was low, autophagy level was low at the basic level, tumor growth curve and tumor quality were not significantly different between WT and 333F groups. However, under the treatment of chemotherapy drugs, IL-6 in tumor microenvironment was increased and promoted the phosphorylation of BECN1-Y333, which led to the increase of autophagy level in WT group compared with 333F group. Autophagy plays a role in tumor protection under stress conditions, so the difference of growth curve and tumor weight between the two groups are significantly increased. A large number of literatures suggest that the same site of post-translational modification may come from different signal regulation. For example, the Y233 site modification of BECN1

can be regulated by both EGFR and Src. Therefore, we also believe that the phosphorylation of BECN1-Y333 may also be regulated by other signal proteins, which is our future research direction.

Autophagy levels in the dissected tumors should be evaluated to confirm the identified role of Y333 in autophagy regulation in xenograft tumor in vivo.

Response: According to your suggestions, we have added the IHC results of LC3B protein as a marker reflecting autophagy levels in revised manuscript.

Clinical relevance

Unclear how the high and low p-BECN1 is defined.

Response: Thanks for the reviewer's suggestion. As suggested, we provided a definition of high and low p-BECN1 levels observed via IHC in tumors in the Methods part. p-BECN1 staining intensity was categorized: no staining as 0, weak as 1, moderate as 2 and strong as 3. The percentage of cells stained was categorized: no positive cells as 0, less than 25% positive cells as 1, 25%-50% positive cells as 2, 50%-75% positive cells as 3 and more than 75% positive cells as 4. We calculated the score of each sample by multiplying the p-BECN1 staining intensity with the percentage scale. For statistical analysis, scores of 0 to 4 were considered low expression and scores of 6 to 12 considered high expression. This part has been added to the revised manuscript.

It remains unclear why Y333-p is associated with significant hazard ratios given that both cell-based assay and xenograft did not show much difference in tumor progression between WT and mutants.

Response: We know that autophagy in tumors usually functions as a protective mechanism when cells are subjected to external stress, such as nutrient deficiency or hypoxia. Herein, in both cell-based assay and xenograft experiments, tumor cells are in a state of adequate nutrition and have not been subjected to adverse factors such as nutrient deficiency or hypoxia, so autophagy generally plays a limited role in proliferation. In the presence of chemotherapeutic drugs, intracellular autophagy plays an important protective role. Therefore, we have found that Y333-BECN1 mediated autophagy promotes cell survival in both xenograft tumors and cells treated with chemotherapy drugs. In terms of clinical treatment, the majority of colorectal cancer patients will receive chemotherapy, so we observe that Y333-p is associated with significant hazard ratios in clinical data.

Combination of JAK2 inhibitor and chemo-drugs, though showing significant effect on cells survival, did not bring further insight into the Y333 mechanisms. Note the downstream umbrella like effect of JAK2 on cell proliferation, differentiation, and apoptosis.

Response: Thank you for this important issue. As you pointed, JAK2 has a number of biological functions in cells, and in fact every protein molecule has many biological functions. In this manuscript, we have demonstrated the role of Y333-BECN1 in chemotherapy resistance through in vitro and in vivo experiments. As an upstream

protein, JAK2 has been clinically developed as a therapeutic drug, we want to examine the role of combination of JAK2 inhibitors and chemotherapy in tumor chemotherapy resistance. Meanwhile, the purpose of this experiment further is to explore the feasibility of that Y333-BECN1 can be used as a biological marker for the chemotherapy sensitivity of clinical patients treated with combination of JAK2 inhibitors and chemotherapy.

Minor Points:

Please provide the link of the multi-database analysis for Tyr identification in the legend of Fig. S4c.

Response: Thank you for your reminding, we provided the link of the multi-database analysis for Tyr identification in the revised legend of **Revised Figure. S4c**.

Fig. 3e, there seems to be more JAK2 WT exogenous expression in WCE than that of K882E mutant; Fig 3F the reduced Tyr phosphorylation of Beclin1 upon JAK2 K882E expression seems not as clear as expected.

Response: In Figure 3e, the expression of internal reference (GAPDH) in later lane was more than that in ahead lane. After quantification, the expression of JAK2 WT exogenous expression in WCE was equal with that of K882E mutant. In addition, we increased exposure time for the band of Tyr phosphorylation of BECN1 and provided it in revised manuscript.

Fig. 4a and 4b could be merged into one graph so that both the basal levels of complex formation and IL-6-mediated increased assembly could be compared in parallel with JAK2 inhibitor.

Response: Thank you for your suggestions. We have repeated the experiment and added it into the revised manuscript.

Fig S6b PARP1 blots need to be repeated. The un-cleaved PARP and cas-3 should be included as controls.

Response: Thank you for your suggestions. According to your advice, we have repeated related experiments and added the bands un-cleaved PARP and cas-3 as controls. The results of western blot were shown in the revised figures.

Fig.6c the marginal region of Normal tissue also showed position staining of p-BECN1, casting doubt on the antigen specificity of this antibody.

Response: To examine the antigen specificity of p-BECN1 antibody, Dot blot and endogenous detection show that the antibody of p- BECN1 Y333 is a specific antibody (Figure 3i, S4e). Of note, normal tissue usually expressed p-BECN1. As shown in Figure 6c, the expression of p-BECN1 in the fibrous stroma tissue outside the mucosa was much more than in normal mucosa tissues. It was not controversial. Immunohistochemical results confirmed that the expression of p-BECN1 indeed in fibrous stroma tissue and tumor tissue was increased and had low levels in normal

mucosal tissues.

Do high levels of p-BECN1 in patient samples correlate with IL-6 too?

Response: IL-6 is normally secreted outside the cell as a cytokine and often enters into blood circulation. In order to confirm the correlation between IL-6 and p-Becn1 in the tumor, a large number of serum samples from tumor patients are needed. The clinical study will be the focus of our later research. It should be noted here that IL-6 has been reported to be highly expressed in cancer patients, including colorectal cancer.

Reviewer #3

Interleukin-6 is tightly tied to tumorigenesis and multiple reports have shown that IL-6 stimulates autophagy in cell culture settings. The mechanism of autophagy activation is however debated. Transcriptional activation of autophagy is rare and evidence for STAT3-mediated control of autophagy is weak.

This manuscript claims that IL-6 activates autophagy not through transcriptional responses mediated by STAT3, but rather through upstream activating Tyrosine phosphorylation by JAK on BECLIN1, a component of a major autophagy-regulating lipid kinase complex. This conveys resistance to anticancer treatment by 5-FU and OXA in vitro. Correlative evidence from colorectal cancer patient samples indicate that phospho-JAK and phospho-Becn1 status predict poorer survival.

The authors first recapitulate earlier results that IL-6 induces autophagy and find that this does depend on Janus Kinase 2 (JAK2) but not the major downstream signalling transducer, STAT3 by knock-out/knock-down and pharmacological means. They move on to show that JAK2 can physically interact with BECN1. Candidate putative direct phosphorylation sites on BECN1 by JAK2 is modelled and tested and Y333 phosphorylation is stimulated by JAK2 activity in vitro and affects the autophagy-regulating VPS34 lipid kinase assembly suggesting a likely direct mechanism for autophagy stimulation. IL-6 mediated resistance to chemotherapy induced cell death is known to depend on autophagy. It is demonstrated that it also depends on the JAK mediated BECN1Y333 phosphorylation in vitro and in a mouse xenograft model. Some correlative patient data suggests that Y333 phospho-BECN1 predicts poor survival suggesting that the JAK2-BECN1-Autophagy axis is relevant to human cancer.

The identification of a direct regulation of autophagy through BECLIN1 by JAK is novel and gives valuable insight in mechanistic control of autophagy of relevance to tumorigenesis. It will be of considerable interest to a wide readership in cell biology, cell signalling, autophagy, immunology and cancer biology.

The manuscript is technically sound and convincing if taken at face value, but source data and statistics, in particular for most Western Blots, are lacking and must be provided too. Co-localization data based on immune fluorescence microscopy is not currently convincing.

The title of the paper “Tumor microenvironment derived IL-6 regulates autophagy and chemotherapy resistance by promoting BECN1 phosphorylation” is not entirely justified as the functional and mechanistic studies are carried out mostly in vitro and the source of IL-6 not addressed. Removing “Tumor microenvironment derived” from the title is an option.

Numerous statements are inaccurate due to poor English and need to be corrected.

Response: We thank the reviewer for recognizing the novelty and importance of our study. We also thank the reviewer for providing the insightful and constructive comments that help strengthen our conclusion further. We follow reviewer’s suggestion and remove “Tumor microenvironment derived” from the title.

1. Using two different cell lines (LoVo and SW48) autophagy flux upon IL-6 stimulation is evaluated using well established autophagy markers: Autophagic GFP-LC3 puncta formation by microscopy, LC3-II (lipidated form) indicating autophagy activation and decrease of the autophagy substrate SQSTM1, and lysosomal release of “free” GFP from GFP-LC3. On the surface, the experiments are sound and carefully performed. However, throughout the paper all Western Blots lack molecular weight markers, number of repeats performed and quantifications + statistics of intensities. The number of independent experiments, source data (Excel files and whole Western blots, not cropped) must be provided. It is impossible to evaluate, with the current images provided, whether correct identification of autophagosomes by transmission electron microscopy has been performed. Please provide better images (higher magnification) in Fig1d and/or supplementary material. I could not find the description of how this quantification was performed in the materials and methods (methods chosen, randomizations, repeats, # of cells, cytoplasmic area covered).

Response: Thank you for putting forward the important suggestion. According to your suggestion, we carefully revised this manuscript, marked molecular weight markers, provided the raw data. In addition, we repeated the experiments of transmission electron microscopy and tried our best to provide high quality images. Meanwhile, we revised the section of materials and methods to provide more detailed information. In order to make the results more reliable, we provided the WB results of IL-6 promoting autophagy in supplements for three times as **Revised Figure.S1j**, and quantitatively analyzed both LC3B and P62 bands in the revised figures. We also provided the original bands in raw data as **Revised Figure.S8**.

2. In Fig1-4 please signify the expected size of LC3 I and II with molecular weight marker/size indicated on the blots.

Response: Thank you for your suggestion, we have marked the molecular weight marker in the revised manuscript.

3. Evaluation of co-localization of JAK and BECN1 by immune fluorescence is not possible at this magnification. Please provide high magnification images using a 63X or 100X objective or even better, super resolution imaging. Provide scale bars.

Response: In the revised manuscript, we repeated this experiment of co-localization of JAK and BECN1 by immune fluorescence. The better images had been provided in our revised figures according to your suggestion.

4. In my opinion, too much stress and space is used at describing the molecular modelling and in Fig3h. It is a useful addition, but based on minimal amounts of experimental data and caution should be exercised to avoid overreach. Statements like “The data shows the modelled structure,....” should not be used.

Response: Thank you for your suggestion, we have revised this part of the statement.

5. Figure 4. Autophagy induction by IL-6 is not evident without quantifications of LC3II vs LC3I, disappearance of SQSTM1 upon stimulation. Provide repeats and quantifications and statistics. Source data, repeats, number of cells, dots and experimental details are lacking for 4g. unless this is provided in files I could not open, this is symptomatic. Please make sure that all data are backed with sufficient experimental detail and source data.

Response: Thank you for your suggestion, the panel of Revised Figure.4g is the quantification results for **Revised Figure.4f**. In the supplementary data, we provide the original data of the full text this time and quantified this data.

6. The term “inactive” unphosphorylated mutants for BECN1 Y333F and Y338F). It is more accurate to use phosphorylation defective. Similarly, the term “active” BECN1 is used for Y333E. As BECN1 is not an enzyme and it is unclear what active/activity means in this setting, the authors are advised to stick to “phospho mimetic”.

Response: Thank you for your suggestion, we have revised the statement in the manuscript.

Minor comments:

Introduction:

«Autophagy, a highly conservative biological behavior that is a lysosomal-dependent degradation process, plays a vital role in cellular protection by recycling degradation

«

This is inaccurate in several ways: rewrite

Response: Thanks for reviewer’s suggestion. We re-write this part as “Autophagy is an intercellular lysosomal-dependent degradation system, which usually engulfs and digests damaged organelles and long-lived proteins and then contributes to survival in response to extracellular and intracellular stress”.

REVIEWERS' COMMENTS

Reviewer #1 (Remarks to the Author):

Authors have answered all of my comments and most of the comments from other reviewers.

Reviewer #2 (Remarks to the Author):

The manuscript has been significantly improved with the supplement of multiple experimental endeavor to further validate the original finding. Most of this reviewers' concerns have been largely addressed that increase the impact of this study.

Reviewer #3 (Remarks to the Author):

Previous remark:

3. Evaluation of co-localization of JAK and BECN1 by immune fluorescence is not possible at this magnification. Please provide high magnification images using a 63X or 100X objective or even better, super resolution imaging. Provide scale bars.

Response: In the revised manuscript, we repeated this experiment of co-localization of JAK and BECN1 by immune fluorescence. The better images had been provided in our revised figures according to your suggestion.

New remark:

The images provided in Fig 3d to assess co-localization of JAK2 and BECN1 are not helpful as they still picture whole cells, and the low resolution and magnification prevents addressing co-localization. Again, images need to be presented where the resolution is sufficiently high by confocal or super-resolution microscopy to assess whether JAK2 and BECN1 is co-localized. This will mean having zoomed in images of only sections of the cytoplasm to reveal internal structures where they may co-localize.

Previous in minor comment section:

Introduction:

«Autophagy, a highly conservative biological behavior that is a lysosomal-dependent degradation process, plays a vital role in cellular protection by recycling degradation»

This is inaccurate in several ways: rewrite

Response: Thanks for reviewer's suggestion. We re-write this part as "Autophagy is an intercellular lysosomal-dependent degradation system, which usually engulfs and digests damaged organelles and long-lived proteins and then contributes to survival in response to extracellular and intracellular stress".

This is still wrong:

New comments:

Replace «intercellular» with «intracellular». The sentence is still awkward and can benefit from rewriting.

Important:

The English in the manuscript is still poor and at times has real consequences in terms of meaning.

For instance, in the abstract it states in line 38 that: "Mechanically, IL-6 triggers the interaction between JAK2 and BECN1, which phosphorylates BECN1 at Y333»

Obviously, "Mechanically" should be "Mechanistically"

The text needs extensive professional English proof reading and editing.

Response to the comments of the reviewers (NCOMMS-20-29331-A)

We would like to thank all the reviewers for recognizing the innovation and importance of our study and for providing constructive comments that in turn help strengthen our study tremendously. We have provided point-by-point answers to address all the reviewers' comments. Importantly, we have provided compelling new experimental data to address the reviewers' concerns.

The point-by-point responses to the reviewers' comments are as followed:

Reviewer #3 (Remarks to the Author):

Previous remark:

3. Evaluation of co-localization of JAK and BECN1 by immune fluorescence is not possible at this magnification. Please provide high magnification images using a 63X or 100X objective or even better, super resolution imaging. Provide scale bars.

Response: In the revised manuscript, we repeated this experiment of co-localization of JAK and BECN1 by immune fluorescence. The better images had been provided in our revised figures according to your suggestion.

New remark:

The images provided in Fig 3d to assess co-localization of JAK2 and BECN1 are not helpful as they still picture whole cells, and the low resolution and magnification prevents addressing co-localization. Again, images need to be presented where the resolution is sufficiently high by confocal or super-resolution microscopy to assess whether JAK2 and BECN1 is co-localized. This will mean having zoomed in images of only sections of the cytoplasm to reveal internal structures where they may co-localize.

Response: In the revised manuscript, the images provided in Fig 3d was actually obtained by using a 63X objective. In the making process of manuscript, the resolution of the picture might be reduced due to inappropriate stretching. To improve the resolution of images provided in Fig 3d, we tried our best to provide the original images in the revised manuscript to better addressing the co-localization between JAK2 and BECN1. We hope the revised manuscript could be acceptable for you.

Previous in minor comment section:

Introduction:

«Autophagy, a highly conservative biological behavior that is a lysosomal-dependent

degradation process, plays a vital role in cellular protection by recycling degradation»
This is inaccurate in several ways: rewrite

Response: Thanks for reviewer's suggestion. We re-write this part as "Autophagy is an intracellular lysosomal-dependent degradation system, which usually engulfs and digests damaged organelles and long-lived proteins and then contributes to survival in response to extracellular and intracellular stress".

This is still wrong:

New comments:

Replace «intercellular» with «intracellular». The sentence is still awkward and can benefit from rewriting.

Response: Thanks for your suggestions. We have corrected this mistake.

Important:

The English in the manuscript is still poor and at times has real consequences in terms of meaning.

For instance, in the abstract it states in line 38 that: "Mechanically, IL-6 triggers the interaction between JAK2 and BECN1, which phosphorylates BECN1 at Y333»

Obviously, "Mechanically" should be "Mechanistically"

The text needs extensive professional English proof reading and editing.

Response: Thanks for your suggestions. We have invited professional experts help polish our article. And we hope the revised manuscript could be acceptable for you.

1. Evaluation of co-localization of JAK and BECN1 by immune fluorescence is not possible at this magnification. Please provide high magnification images using a 63X or 100X objective or even better, super resolution imaging. Provide scale bars.

Response: In the revised manuscript, we repeated this experiment of co-localization of JAK and BECN1 by immune fluorescence. The better images had been provided in manuscript according to your suggestion.

2. In my opinion, too much stress and space is used at describing the molecular modelling and in Fig3h. It is a useful addition, but based on minimal amounts of experimental data and caution should be exercised to avoid overreach. Statements like “The data shows the modelled structure,....” should not be used.

Response: Thank you for your suggestion, we have revised this part of the statement.

Minor comments:

Introduction:

«Autophagy, a highly conservative biological behavior that is a lysosomal-dependent degradation process, plays a vital role in cellular protection by recycling degradation
«

This is inaccurate in several ways: rewrite

Response: Thanks for reviewer’s suggestion. We re-write this part as “Autophagy is an intercellular lysosomal-dependent degradation system, which usually engulfs and digests damaged organelles and long-lived proteins and then contributes to survival in response to extracellular and intracellular stress”.